# Enhancing Reasoning in Large Language Models via Entropy-Aware Self-Evolution

## Abstract

Large language models (LLMs) have exhibited remarkable reasoning capabilities. However, when self-evolution frameworks are employed to further enhance these models, a key challenge lies in balancing correctness, which ensures reliable supervision, and exploration, which promotes diverse reasoning trajectories. To address this dilemma, we propose an **entropy-aware self-evolution framework** that integrates verifier feedback with both sequence-level and token-level entropy. Our approach incorporates two key strategies: (i) *high-entropy selection* of verified trajectories to provide informative yet reliable signals; and (ii) *entropy-aware rethinking*, which revisits uncertain reasoning steps to uncover alternative solutions. Theoretically, we establish the connection between entropy and the expected supervised fine-tuning loss, showing that high-entropy trajectories yield stronger learning signals. Empirically, experiments across multiple reasoning benchmarks demonstrate that our framework consistently improves both reliability and exploratory capacity over strong baselines. With the assistance of the proposed framework, InternLM2.5-1.8B achieves an improvement of **8.27%** and surpasses the strong baseline by **1.82%** on the GSM8K task, as measured by $Pass@16$. Our results highlight entropy as a principled driver of self-improvement, enabling LLMs to evolve toward models that are not only more accurate but also more exploratory.

## 1 Introduction

Large language models (LLMs) have shown impressive reasoning capabilities across tasks such as mathematical problem solving, code generation, and scientific discovery (OpenAI, 2024; DeepSeek-AI, 2025; Zhu et al., 2025). Despite these successes, traditional training methods often rely on static datasets and may not fully exploit the models' potential for iterative improvement. A growing trend, known as self-evolution, addresses this by generating new training trajectories and fine-tuning models iteratively on them (Wang et al., 2022; Xu et al., 2025; Zhou et al., 2025). While this approach supports scalable iterative self-improvement, it faces a fundamental dilemma: models must balance **correctness** (ensuring generated trajectories are valid and high-quality) with **exploration** (encouraging diverse and novel reasoning paths that might reveal new insights).

Existing approaches to self evolution typically lean towards one side of this trade-off. Verifier-based or reinforcement learning with verifiable rewards (RLVR) methods (Lambert et al., 2025; Shao et al., 2024) prioritize correctness by filtering out invalid trajectories and aligning models with reliable supervision. However, these methods often bias learning toward low-perplexity, deterministic reasoning paths, thereby diminishing exploration and leading to convergent behaviors (Yue et al., 2025). Conversely, exploration-driven strategies based on entropy, perplexity, or trial-and-error sampling (Wang et al., 2025b; Li et al., 2025; Deng et al., 2025) encourage diversity, but correctness is not guaranteed, producing noisy or misleading training signals. Consequently, despite significant progress, current self-evolution frameworks struggle to balance correctness and exploration effectively.

To address the correctness–exploration trade-off, we present an entropy-aware self-evolution framework. Our key insight is that verified high-entropy trajectories not only furnish reliable supervision but also, by leveraging their intrinsic uncertainty, illuminate alternative reasoning paths that warrant exploration. By exploiting entropy at both the sequence and token level, and integrating verifier

feedback, our framework achieves a principled balance between correctness—providing dependable learning signals—and exploration—enabling diverse and informative data generation. Specifically, the framework employs two complementary strategies: (i) **High-Entropy Selection**, which prioritizes trajectories with high uncertainty yet verified correctness to supply both informative and reliable training signals; and (ii) **Entropy-Aware Revisiting of Reasoning Steps**, which identifies high-uncertainty reasoning positions for truncation and regeneration, uncovering alternative solutions and promoting exploratory reasoning. Experiments across different models and tasks demonstrate the superiority of our proposed method, surpassing the strong baseline by **1.44%-5.52%** at average performance on four math reasoning tasks. Our contributions are as follows:

- We propose a novel high-entropy trajectory selection strategy that balances correctness and exploration, addressing a key limitation of prior low-perplexity–biased frameworks.
- We introduce an entropy-aware rethinking mechanism that revisits uncertain reasoning steps, systematically enriching solution diversity while preserving reliability.
- We provide both theoretical analysis, establishing the link between sequence-level entropy and expected supervised fine-tuning loss, and extensive empirical validation on reasoning benchmarks, demonstrating that our framework consistently improves both reliability and exploratory capacity compared to strong baselines.

## 2 RELATED WORK

**Self-Evolution with Data Synthesis and Selection.** Existing self-evolution approaches for LLMs have explored a variety of strategies for data synthesis and selection. Prior work on data synthesis for self-evolution has relied on heuristic filtering (Wang et al., 2022), confidence-based ranking (Huang et al., 2023), or similarity measures (Chen et al., 2024), while others incorporate external verifiers or interactive environments (Xu et al., 2025; Zhou et al., 2025). Although these strategies improve correctness, they often sacrifice data diversity, leading to convergent trajectories in later training stages. Recent uncertainty-aware approaches leverage entropy (Wang et al., 2025b), perplexity (Li et al., 2025), or exploration-driven sampling (Deng et al., 2025) to encourage diversity, but lack fine-grained utilization of trajectory entropy dynamics. In contrast, our method combines an external verifier with both trajectory-level and token-level entropy guidance, ensuring correctness while systematically enriching diversity and exploration, thus achieving a balanced and robust self-evolution process.

**Reinforcement Learning using Verifiable Rewards.** With the increasing adoption of reinforcement learning in LLM training, Reinforcement Learning with Verifiable Rewards (RLVR) (Lambert et al., 2025) has emerged as a promising paradigm for enhancing reasoning in LLMs. Similar to our study, RLVR can be viewed as a self-evolution framework that integrates external verifiers. Notably, models such as OpenAI o1(OpenAI, 2024) and DeepSeek-R1(DeepSeek-AI, 2025) exemplify the effectiveness of this approach. In particular, DeepSeek-R1 employs the GRPO (Shao et al., 2024), which eliminates reliance on a reward model and has inspired a range of extensions such as DAPO(Yu et al., 2025) and VAPO(Yue et al., 2025). However, recent analyses indicate several limitations: post-RL models often exhibit reduced exploration compared to their base counterparts(Yue et al., 2025); and correct rewards may still be entangled with erroneous reasoning steps, leading to noisy training signals(Yee et al., 2024; Wan et al., 2025; Wen et al., 2025). Similar to some works on RL with an entropy perspective(Wang et al., 2025a; Cheng et al., 2025), our method leverages entropy-driven self-evolution to preserve exploration ability, operates effectively in domain-specific tasks without requiring long nature language CoTs, and employs a robust external verifier to ensure correctness, thereby avoiding reinforcement of spurious reasoning.

## 3 METHOD

As shown in Figure 1, We propose an entropy-aware self-evolution framework for LLMs, composed of three stages: (1) **Trajectory Exploration** — generating candidate reasoning trajectories to probe the task space, (2) **Trajectory Rethinking** — revisiting uncertain reasoning steps to diversify problem-solving paths, and (3) **Trajectory Selection** — curating informative trajectories to enhance both training signal and model exploration ability.

The central advantage of this design lies in its explicit focus on *high-entropy samples*, which are indicative of epistemic uncertainty and exploratory potential. By prioritizing such samples and leveraging verifier feedback, our framework not only improves data quality but also systematically encourages the model to explore alternative reasoning paths. The pipeline is iterated for $I$ steps, starting with a base model $\pi_0$ at iteration $i = 0$.

### 3.1 Entropy Measures for Model Trajectories.

We quantify uncertainty in model-generated trajectories using *token-level* and *sequence-level* entropy.

**Local uncertainty**: We utilize the token-level entropy to capture local uncertainty and inform *high-entropy truncation and revisiting* during trajectory refinement. Formally, the token-level entropy at position $t$ is defined as

$$H_t = -\sum_{i=1}^{V} p_\theta(v_i|\boldsymbol{y}_{<t}, \boldsymbol{x}) \log p_\theta(v_i|\boldsymbol{y}_{<t}, \boldsymbol{x}), \tag{1}$$

where $p_\theta(v_i|\boldsymbol{y}_{<t}, \boldsymbol{x})$ is the model's predictive probability for token $v_i$ given prefix $\boldsymbol{y}_{<t}$ and input $\boldsymbol{x}$. A low $H_t$ indicates that the model's predictions are concentrated on a small set of tokens, reflecting high confidence, while high $H_t$ reflects multiple plausible alternatives, creating branching points that can decisively influence the trajectory.

**Global uncertainty**: We utilize the sequence-level entropy that aggregates token-level uncertainties to measure global unpredictability of a trajectory $\boldsymbol{y} = (y_1, \ldots, y_T)$:

$$H_{\text{seq}}(\boldsymbol{y} \mid \boldsymbol{x}) = \frac{1}{T} \sum_{t=1}^{T} H_t. \tag{2}$$

Trajectories with high $H_{\text{seq}}$ contain multiple positions with substantial uncertainty, indicating both higher exploratory potential and richer information content. Conversely, low $H_{\text{seq}}$ trajectories correspond to more deterministic generations. Sequence-level entropy thus provides an effective criterion for selecting uncertainty and exploratory trajectories in supervised fine-tuning (SFT).

In out framework, token-level entropy identifies critical positions for trajectory refinement, while sequence-level entropy selects high-information trajectories for SFT. By leveraging both, the model benefits from trajectories that are both exploratory and informative, thereby enhancing the task-specific performance of LLMs.

### 3.2 Trajectory Exploration

We start by broadly exploring the solution space, allowing the model to generate candidate trajectories while quantifying their uncertainty. Let $\mathcal{D}$ denote a task-specific dataset comprising instruction-answer pairs $(\boldsymbol{x}, a)$. At iteration $i$, the current model $\pi_i$ generates $K$ trajectories for each input $\boldsymbol{x}$: $\{\boldsymbol{y}_k\}_{k=1}^{K} \sim \pi_i(\cdot \mid \boldsymbol{x})$. For each trajectory $\boldsymbol{y}_k$, we compute its sequence-level entropy: $h_k = H_{\text{seq}}(\boldsymbol{y}_k \mid \boldsymbol{x})$. Each trajectory is then verified by an external checker (Xu et al., 2025), yielding a correctness label: $r_k = \text{validator}(\boldsymbol{y}_k, a), r_k \in \{0, 1\}$. The final quadruple is stored as $T_k = (\boldsymbol{x}, \boldsymbol{y}_k, h_k, r_k)$. All positively verified trajectories are aggregated into the *exploration pool*:

$$\mathcal{P}_i^+ = \{ T_k \mid r_k = 1 \}_{k=1}^{K} \cup \mathcal{P}_{i-1}^+, \quad \mathcal{P}_{-1}^+ = \varnothing. \tag{3}$$

This pool serves as the foundation for subsequent trajectory selection.

### 3.3 Trajectory Rethinking

Prior work (Wang et al., 2025c; Gao et al., 2025) emphasizes that medium-difficulty and uncertain samples play a crucial role in self-training. To better exploit such informative cases, we introduce *trajectory rethinking*, which revisits high-entropy reasoning steps to encourage exploration of alternative solutions.

From the verified trajectories of this iteration $\{ T_k \mid r_k = 1 \}_{k=1}^{K}$, we select the positive trajectory with the highest sequence-level entropy: $\boldsymbol{y}^\star = \arg\max_{\boldsymbol{y}_k \in \mathcal{P}_i^+} H_{\text{seq}}(\boldsymbol{y}_k \mid \boldsymbol{x})$. Let $T$ be the length

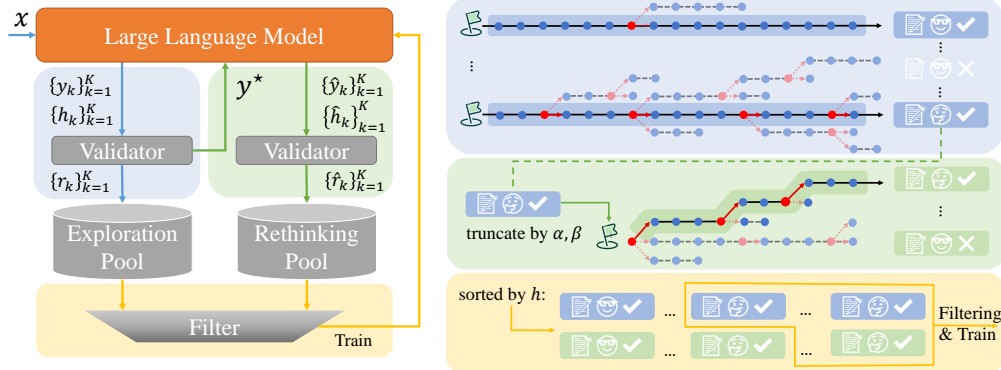

Figure 1: (**Left**) Pipeline shows our entropy-aware self-evolution framework. (**Right**) Three stages for the framework. Three background colors in the left—blue, green, and yellow—indicate the same stages as those in the right from top to bottom.The trajectory exploration stage, highlighted in blue, illustrates how the model explores and verifies candidate trajectories, as detailed in Section 3.2;The trajectory rethinking stage, highlighted in green, illustrates how we leverage the explored correct trajectories to truncate and regenerate, as detailed in Section 3.3.The trajectory selection stage, highlighted in yellow, selects highly exploratory and informative trajectories to enhance the model's capabilities, as detailed in Section 3.3. Through repeated iterations of this framework, we construct a set of trajectories that are both reliable and exploratory, which facilitates the enhancement of the model's task execution and exploratory capabilities.The three stages progressively transform raw trajectories into reliable yet diverse supervision signals.

of $\boldsymbol{y}^\star$. Token-level entropies $H_t$ are used to identify uncertain positions. With hyperparameters $\alpha \in (0,1)$ (fraction of top-entropy tokens) and $\beta \in (0,1)$ (maximum truncation ratio), we define the candidate set:

$$\mathcal{I} = \{t \mid t \leq \lfloor \beta T \rfloor,\ y_t^\star \in \text{Top}_\alpha(H_t)\}. \tag{4}$$

We then sample a truncation point: $\tau \sim \text{Uniform}(\mathcal{I})$, and obtain the truncated prefix: $\boldsymbol{y}_{\leq \tau}^\star = (y_1^\star, \ldots, y_\tau^\star)$. Conditioned on $(\boldsymbol{x}, \boldsymbol{y}_{\leq \tau}^\star)$, the model generates $K$ continuations: $\{\tilde{\boldsymbol{y}}_{k,>\tau}\}_{k=1}^K \sim \pi_i(\cdot \mid \boldsymbol{x}, \boldsymbol{y}_{\leq \tau}^\star)$, which are concatenated with the prefix to form *rethought trajectories*: $\{\tilde{\boldsymbol{y}}_k\}_{k=1}^K = \{\boldsymbol{y}_{\leq \tau}^\star \oplus \tilde{\boldsymbol{y}}_{k,>\tau}\}_{k=1}^K$. All rethought trajectories are verified, and positives are aggregated into the *rethinking pool*:

$$\tilde{\mathcal{P}}_i^+ = \left\{ \tilde{T}_k = (\boldsymbol{x}, \tilde{\boldsymbol{y}}_k, \tilde{h}_k, \tilde{r}_k) \mid \tilde{r}_k = 1 \right\}_{k=1}^K \cup \tilde{\mathcal{P}}_{i-1}^+, \quad \tilde{\mathcal{P}}_{-1}^+ = \varnothing. \tag{5}$$

When no positively verified samples exist, we apply the procedure to the negative trajectory with the highest sequence-level entropy, so that high-entropy trajectories, regardless of their correctness, continue to drive exploration of alternative reasoning paths.

## 3.4 Trajectory Selection

During the self-evolution process, the contributions of different generated trajectories to model learning vary significantly. To maximize the utility of limited training resources, it is necessary to select trajectories that are both exploratory and information-rich from a large pool of candidates. The trajectory selection stage aims to aggregate and identify these critical trajectories to enhance the model's learning. By emphasizing high-entropy trajectories, this selection process encourages the model to explore uncertain regions of the solution space, thereby acquiring a more comprehensive reasoning experience.

Specifically, we rank both $\mathcal{P}_i^+$ and $\tilde{\mathcal{P}}_i^+$ in descending order of sequence-level entropy, obtaining $\mathcal{R}_i^+$ and $\tilde{\mathcal{R}}_i^+$. From these, we select the top-$N$ trajectories from the exploration pool:

$$\mathcal{T}_1 = \left\{ (\boldsymbol{x}, y_n) \mid n \leq \min\left(N, |\mathcal{R}_i^+|\right),\ T_n \in \mathcal{R}_i^+ \right\}. \tag{6}$$

If $|\mathcal{T}_1| < N$, we fill the remainder from the rethinking pool:

$$\mathcal{T}_2 = \left\{ (\boldsymbol{x}, \tilde{\boldsymbol{y}}_n) \mid n \leq \min\left(N - |\mathcal{T}_1|, |\tilde{\mathcal{R}}_i^+|\right),\ \tilde{T}_n \in \tilde{\mathcal{R}}_i^+ \right\}. \tag{7}$$

**Supervised fine-tuning on the filtering trajectories.** We fine-tune the model $\pi_0$ on $\mathcal{T} = \mathcal{T}_1 \cup \mathcal{T}_2$ using maximum likelihood estimation (MLE) also known as the cross-entropy loss $\mathcal{L}_{CE}$ to get next-iteration model $\pi_{i+1}$,

$$\mathcal{L}_{\text{CE}} = - \sum_{(\boldsymbol{x},\boldsymbol{y}) \sim \mathcal{T}_1 \cup \mathcal{T}_2} \log p_\theta(\boldsymbol{y} \mid \boldsymbol{x}). \tag{8}$$

### 3.5 Analysis of the Relationship Between Entropy and the Expected Supervised Loss

The defination of cross-entropy loss for SFT on one self-generated trajectory $\boldsymbol{y}$ is

$$\mathcal{L}_{CE}(\boldsymbol{y}|\boldsymbol{x}) = - \sum_{t=1}^{T} \log p_\theta(y_t \mid \boldsymbol{y}_{<t}, \boldsymbol{x}). \tag{9}$$

Its expectation over trajectories sampled from the model $\pi_\theta(\cdot|\boldsymbol{x})$ can be expressed as

$$\mathbb{E}_{\boldsymbol{y} \sim \pi_\theta(\cdot|\boldsymbol{x})}[\mathcal{L}_{CE}(\boldsymbol{y}|\boldsymbol{x})] = - \sum_{t=1}^{T} \mathbb{E}_{\boldsymbol{y} \sim \pi_\theta(\cdot|\boldsymbol{x})}[\log p_\theta(y_t \mid \boldsymbol{y}_{<t}, \boldsymbol{x})] \tag{10}$$

$$= \sum_{t=1}^{T} \mathbb{E}_{\boldsymbol{y}_{<t} \sim \pi_\theta(\cdot|\boldsymbol{x})}[H_t] \tag{11}$$

$$= T \cdot \mathbb{E}_{\boldsymbol{y} \sim \pi_\theta(\cdot|\boldsymbol{x})}[H_{\text{seq}}(\boldsymbol{y} \mid \boldsymbol{x})], \tag{12}$$

where the second equality follows from the definition of token-level entropy and the last equality from sequence-level entropy. This relationship shows that higher-entropy trajectories induce larger expected loss, producing stronger gradients and richer learning signals. Additionally, we discuss the theoretical analysis of entropy as an exploration-enhancing signal, beyond its role in training value, in the Appendix D.3.

Overall, our method combines verifier guidance with entropy-aware trajectory selection. By explicitly exploiting high-entropy samples for both exploration and augmentation, the framework not only ensures training quality but also enhances the model's ability to explore and generalize across uncertain reasoning pathways. Through iterative self-evolution, the model progressively improves its task-specific reasoning performance.

## 4 Experiments

### 4.1 Experimental Setup

**Datasets.** We evaluate the proposed framework on math reasoning tasks, using a Python executor as the validator. Reasoning tasks include: GSM8K(Cobbe et al., 2021), MATH(Hendrycks et al., 2021), GSM-Hard(Gao et al., 2023), SVAMP(Patel et al., 2021), and AsDiv(Miao et al., 2020). The training split of GSM8K, along with randomly selected samples from MATH, is used to construct the dataset witn 13,492 samples for self-evolution. The test splits of GSM8K, GSM-Hard, SVAMP, and AsDiv are reserved for evaluation. In order to make use of the validator, we prompt the LLM to generate reasoning path with the format of executable python code.

**Training Details.** We use Qwen2.5-Instruct(Yang et al., 2024; Qwen, 2024), Llama3.2(Grattafiori et al., 2024; Meta, 2024) and InternLM-2.5(Cai et al., 2024) models for evaluation. At the first iteration, we utilize few-shot prompting to instruct the model to generate training samples as a cold start. The few-shot numbers for math reasoning tasks are set to 3. At each evolution iteration, the candidate trajectory size $K$ is set to 5. The total iteration number $I$ is set to 10 for InternLM2.5-1.8B, 7 for Llama3.2-1B and 7 for Qwen2.5-Instruct-1.5B. The top-$N$ for trajectory augmentation is set to 10. Otherwise, we make use of the negative trajectories the same as the baseline (Xu et al., 2025). All the self-evolution experiments are implemented on $4\times$RTX3090 of 24GB VRAM.

## 4.2 MAIN RESULTS

Table 1 summarizes the evaluation results across four mathematical reasoning benchmarks. For reference, we include a few-shot baseline, while all other evaluations are conducted under the zero-shot setting. To ensure fairness, all experiments adopt a consistent sampling strategy with top-$p = 0.95$ and temperature $= 0.6$. We further compare our approach with the ENVISIONS framework (Xu et al., 2025) under identical conditions and the main differences with ENVISIONS and the reason why we chose it as the baseline are discussed in the Appendix E.5 . To evaluate both accuracy and exploratory capacity, we use $Pass@K$ as the primary metric, as it reflects the model's ability to produce correct solutions under multiple sampled attempts.

**Overall Performance Improvements.** Our method delivers substantial improvements over the base models and consistently outperforms ENVISIONS, as shown in Tabel 1. On the held-in task GSM8K, InternLM2.5-1.8B achieves a remarkable 8.27% gain at $Pass@16$. Compared with ENVISIONS, our method yields improvements of 1.82% and 4.39% at $Pass@16$ and $Pass@128$, respectively, along with an average performance gain of 2.57% when $K$ ranges from 16 to 256. These results indicate that our approach not only strengthens task execution accuracy relative to the base models, but also enhances exploratory capacity when compared to existing frameworks.

**Generalization to Held-out Benchmarks.** To examine generalization, we conduct evaluations on GSM-Hard, AsDiv, and SVAMP (Table 1). Consistent with the observations on GSM8K, our method achieves clear gains over the base models and surpasses ENVISIONS on GSM-Hard and AsDiv. On GSM-Hard, InternLM2.5-1.8B improves by 7.21% and delivers an additional 1.44% average gain compared with ENVISIONS. On SVAMP and AsDiv, our method outperforms the baseline by 5.52% and 5.51% in average performance, respectively. These results demonstrate the strong generalization ability of our framework across diverse reasoning benchmarks. Moreover, on SVAMP, which is a relatively simple benchmark, InternLM2.5-1.8B already matches or exceeds the performance of self-evolution variants under few-shot settings. In contrast, our method better preserves the exploratory capacity of the base models, whereas ENVISIONS exhibits a noticeable decline.

**Generalization to Various Backbones.** We also compare our method with ENVISIONS on Llama3.2-1B and Qwen2.5-Instruct-1.5B. As shown in Figure 2, our method consistently outperforms ENVISIONS across tasks and backbones. Significantly, as illustrated in Figure 3, the performance improvements become more pronounced at larger $K$, highlighting that our evolutionary strategy effectively enhances the ability of models to explore diverse solution trajectories.

Table 1: Math Reasoning results of InternLM2.5-1.8B on four tasks.

| | GSM8K | | | GSM-Hard | | | SVAMP | | | AsDiv | | |
|---|---|---|---|---|---|---|---|---|---|---|---|---|
| | $Pass@16$ | $Pass@128$ | $Avg$ | $Pass@16$ | $Pass@256$ | $Avg$ | $Pass@16$ | $Pass@256$ | $Avg$ | $Pass@16$ | $Pass@128$ | $Avg$ |
| *InternLM2.5-1.8B* | | | | | | | | | | | | |
| Few-shot | 63.53 | 84.00 | 73.73 | 52.84 | 74.68 | 60.93 | **84.30** | **95.70** | **89.52** | 76.01 | 84.68 | 80.00 |
| ENVISIONS | 69.98 | 80.67 | 75.07 | 59.36 | 71.19 | 64.20 | 79.50 | 88.20 | 83.01 | 72.97 | 78.44 | 75.68 |
| Ours | **71.80** | **85.06** | **77.64** | **60.05** | **75.21** | **65.64** | 83.90 | 95.10 | 88.53 | **77.61** | **85.42** | **81.19** |
| $\Delta$ | +1.82 | +4.39 | +2.57 | +0.68 | +4.02 | +1.44 | +4.40 | +6.90 | +5.52 | +4.64 | +6.98 | +5.51 |

## 4.3 EVOLUTION PROGRESS FOR SELF-EVOLUTION FRAMEWORKS

As illustrated in Figure 4(**Left**), the iterative evolution curves of the self-training frameworks with InternLM2.5-1.8B as the LLM, demonstrate the progression of performance improvement. Compared with the ENVISIONS method, our framework exhibits a more pronounced performance improvement. Notably, while the performance of ENVISIONS tends to plateau after the fourth iteration, our method not only achieves superior results but also shows continued potential for further improvement. From Figure 4 (**Right**), it can be observed that under our framework, both the mean and variance of sequence-level entropy in the training dataset increase as the number of self-evolution iterations grows, exhibiting a trend in sharp contrast to that of the ENVISIONS method.

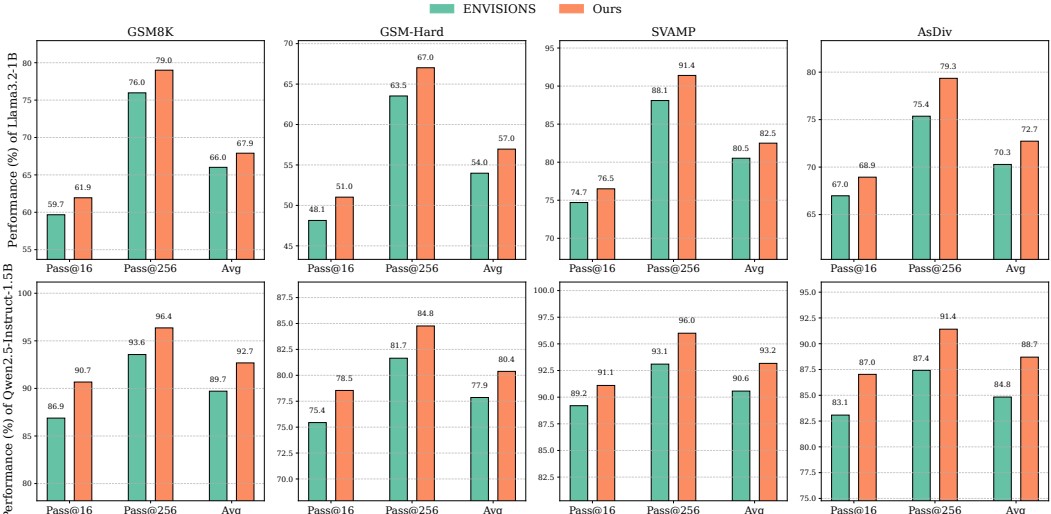

Figure 2: Math Reasoning evaluation of the Llama3.2-1B and Qwen2.5-Instruct-1.5B on the four tasks, compared with the existing method.

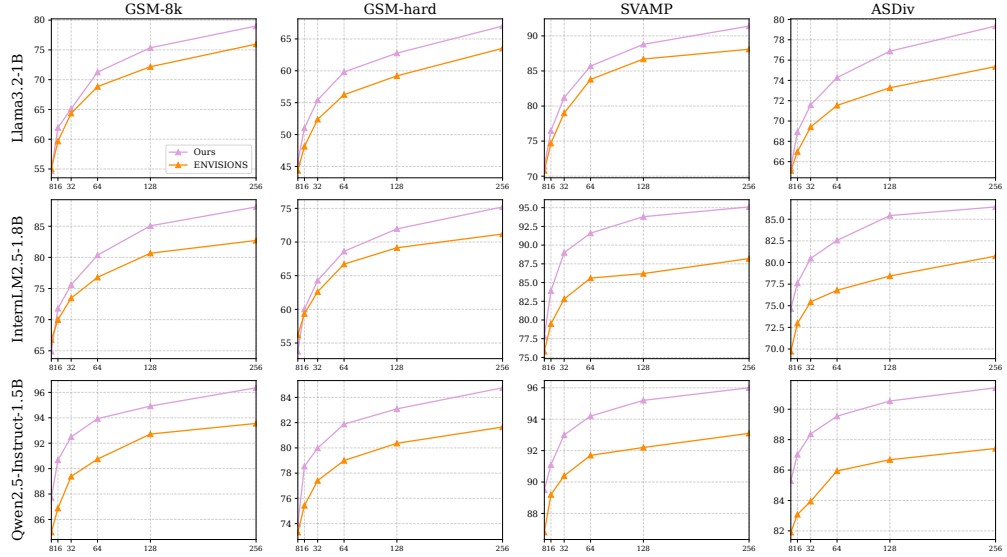

Figure 3: $Pass@K$ performance of the LLMs with different self-evolution frameworks. The horizontal axis denotes $K$ ranging from 8 to 256, and the vertical axis shows the corresponding $Pass@K$ accuracy on the benchmarks.

# 5 ANALYSIS

## 5.1 ABLATION STUDIES

**Experiment Setups** To disentangle the contribution of each module in our framework, we conduct ablation studies over four configurations. All settings use a maximum of $N = 10$ samples for SFT and $I = 10$ iterations for self-evolution. For a compute-matched comparison, the **Selection Only** variant sets $K = 10$, compensating for the absence of the rethink/refine stage (self-refine in ENVISIONS) so that it produces the same number of trajectories per iteration as the two-stage variants that use $K = 5$. For the **Rethink Only** variant, we uniformly sample $N$ trajectories from the candidate pool without entropy-based selection when constructing the SFT dataset. We evaluate the variant self-evolution methods using InternLM2.5-1.8B on the 1k-sample subset of the full dataset.

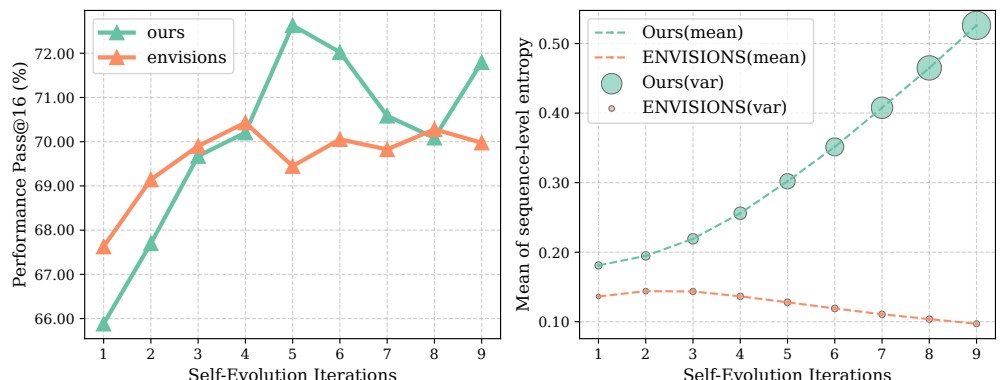

Figure 4: (**Left**) Performance evolution of two frameworks on InternLM-2.5-1.8B model. (**Right**) Mean and variance of sequence-level entropy of the SFT training datas for each evolution.

Table 2: Ablation results on GSM8K using InternLM2.5-1.8B trained on a 1k-sample subset. All variants are compute-matched with respect to total generated trajectories.

| Method Variant | Pass@16 (%) |
|---|---|
| Full Method (Selection + Rethinking) | **53.68** |
| Exploration + Selection Only | 49.12 |
| Exploration + Rethinking Only | 50.42 |
| ENVISIONS | 50.27 |

**Component Ablation Studies**  Table 2 summarizes the results on GSM8K. Both partial variants—**Selection Only** and **Rethink Only**—provide moderate improvements, demonstrating that each component independently contributes to performance. The full method, which combines exploration-driven selection with the subsequent rethinking stage, yields a substantially larger gain, achieving a Pass@16 of 53.68%. This confirms that the two components are complementary: selection biases the model toward higher-quality trajectories, while the rethinking stage further increases both the quantity and quality of these trajectories. Compared to ENVISIONS, our full framework achieves a 3.4% improvement, validating the effectiveness of our exploration and rethinking design.

**Comparison Between Selection Strategies.**  To evaluate the effectiveness of high-entropy selection, we compare three trajectory selection strategies: (i) *High-Entropy*, which selects the top-$N$ highest-entropy trajectories; (ii) *Low-Entropy*, which selects the top-$N$ lowest-entropy trajectories; and (iii) *Entropy-free*, which randomly samples $N$ trajectories from the set of correct trajectories. We evaluate these variants on the 1k-sample subset of the full dataset using InternLM2.5-1.8B, following the same experimental setup described earlier in this section. The results, summarized in Table 3, show that High-Entropy selection achieves the best performance (53.68%), random selection yields moderate performance (50.42), and Low-Entropy selection performs the worst (48.78). This contrast clearly demonstrates that high-entropy trajectories provide more diverse decision forks, enabling more effective exploration of the model's potential and reasoning space during the self-evolution process.

| Selection Strategy | Pass@16(%) |
|---|---|
| High-Entropy | 53.68 |
| Low-Entropy | 48.78 |
| Entropy-free (Random) | 50.42 |

Table 3: Comparison of different trajectory selection strategies.

## 5.2 HIGH-ENTROPY SELECTION ENHANCES TRAINING INFORMATION AND TRAJECTORY DIVERSITY

To further investigate the effect of out high-entropy selection strategy, we analyze the distribution of similarity scores and negative log probability of the selected trajectories for the last self-evolution iteration of three models.

The similarity score quantifies the alignment among generated trajectories, with higher values indicating greater overlap and lower values reflecting higher diversity. Formally, given a set of $n$ trajectories $(t_1, t_2, \ldots, t_n)$ corresponding to the same problem, we obtain their embeddings $\{\mathbf{e}_i\}_{i=1}^n$ from a pretrained embedding model $f(\cdot)$(Zhang et al., 2025). The similarity score is computed as

$$\text{Sim} = \frac{1}{n(n-1)} \sum_{i=1}^{n} \sum_{\substack{j=1 \\ j \neq i}}^{n} \langle f(t_i^q), f(t_j^d) \rangle$$

where $f(t_i^q)$ and $f(t_j^c)$ denote query-style and candidate-style embeddings of trajectory $t$, and $\langle \cdot, \cdot \rangle$ denotes the inner product. See Appendix C for more details.

As shown in the top row of Figure 5, our method produces a wider distribution of similarity scores with a noticeable shift toward lower values compared to ENVISIONS, indicating that high-entropy selection promotes greater trajectory diversity. The trajectory examples presented in the Appendix E.5 across different iterations further illustrate the diversity gains introduced by our selection strategy. Meanwhile, the bottom row reveals that our approach selects trajectories with higher negative log probabilities, implying that the chosen samples carry more informative signals rather than being restricted to high-confidence outputs. Our analysis of computational efficiency in the Appendix C further confirms that providing richer training signals leads to improved training efficiency. Overall, these results demonstrate that high-entropy selection enhances both the information content and the diversity of the training data, which are crucial for improving the expertise and generalization capability of LLMs in self-evolution frameworks.

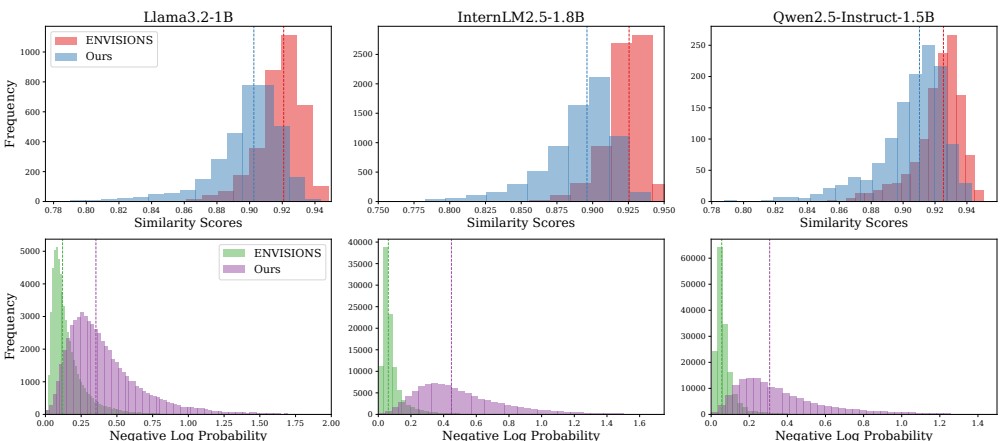

Figure 5: Histogram of Similarity Scores and Negative Log Probability of the trajectories selected for the last self-evolution iteration. The dashed lines in the figures denote the median.

## 5.3 THE ROLE OF TRAJECTORY RETHINKING IN SELF-EVOLUTION.

To analyze the role of the *Trajectory Rethinking* stage within our framework, we conduct an in-depth investigation from three perspectives. First, we evaluate its impact on reasoning performance. Specifically, we evaluate InternLM2.5-1.8B on GSM8K under a 1k-sample training budget, comparing performance with and without the *Trajectory Rethinking* stage. As shown in Figure 6 (Left), incorporating Trajectory Rethink consistently boosts Pass@16 across iterations, indicating a clear and stable improvement. In contrast, the variant without this stage—relying solely on *Trajectory Exploration*—exhibits noticeably weaker performance.

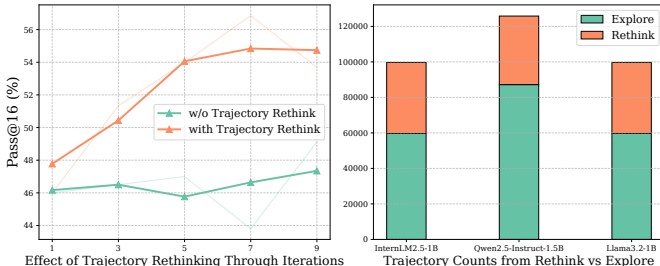

Figure 6: **Analysis of Trajectory Rethinking in self-evolution.** (**Left**) Performance across iterations: Incorporating the rethinking stage consistently outperforms the variant without rethinking at every iteration. (**Middle**) Trajectory Counts: Rethink and explore complement each other across different base models, leading to an increase in effective training samples. (**Right**) High-Entropy Tokens: The frequent occurrence of truncated tokens with high entropy indicates that rethinking mitigates uncertainty and enhances trajectory diversity.

Moreover, we examine the contribution of *Trajectory Rethink* to trajectory diversity. Figure 6 (**Middle**) shows that this strategy accounts for more than one-third of the training trajectories generated during the evolution process, substantially enriching the diversity of the training data. This indicates that rethink contributes significantly to the breadth of explored reasoning paths.

Finally, we analyze the linguistic patterns associated with rethink. We visualize the most frequent truncated tokens with high entropy, as shown in Figure 6 (**Right**). Words such as *"because"*, *"since"*, and *"then"* often determine the direction of reasoning. Truncating trajectories at these critical tokens enables the model to rethink from pivotal decision forks, thereby facilitating more flexible and diverse reasoning. These analyses demonstrate that Trajectory Rethink is a crucial component of our self-evolution framework. It enhances the diversity of reasoning trajectories and encourages re-exploration from meaningful reasoning pivots, ultimately leading to richer and more informative training signals, particularly beneficial for challenging reasoning tasks.

## 6 CONCLUSION

We propose an entropy-aware self-evolution framework that enhances reasoning in large language models by strategically leveraging uncertainty to balance correctness and exploration. Integrating verifier feedback with sequence-level and token-level entropy, our method prioritizes high-entropy yet verified trajectories for training, ensuring reliable supervision while actively promoting diverse reasoning paths. Theoretical analysis shows that such trajectories yield stronger learning signals due to their higher expected loss, enabling more effective fine-tuning. Empirically, our approach achieves significant gains across multiple reasoning benchmarks. Notably, InternLM2.5-1.8B improves by **8.27%** on GSM8K at Pass@16 and surpasses the strong ENVISIONS baseline by **4.39%** at $Pass@128$, with consistent gains on held-out tasks like GSM-Hard, SVAMP and AsDiv. Critically, performance improvements grow with larger sampling budgets, confirming enhanced exploration without sacrificing accuracy.

**Limitation** Our experiments are limited to models up to 1.8B parameters due to computational constraints; scaling to larger architectures (e.g., 7B+) remains untested. The framework's reliance on executable verifiers also restricts current applicability to math/code domains. Future work will address efficiency, entropy approximation, and extension to semantic reasoning tasks.

In summary, our entropy-aware self-evolution framework offers a principled, theoretically grounded, and empirically validated approach to enhancing both the reliability and exploratory capacity of LLMs. By treating uncertainty not as noise to be suppressed but as signal to be harnessed, we enable models to evolve into more capable, flexible, and robust reasoners.

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

## LLM USAGE

We used large language models (LLMs) as auxiliary tools for writing assistance and language polishing. Specifically, LLMs were employed to improve readability, grammar, and presentation of the text. All research ideas, experimental designs, and scientific contributions are entirely the work of the authors. The authors take full responsibility for the content of this paper.

## A   TRAINING DETAILS

The SFT training in our framework and baselines is conducted on 4×RTX3090 with a maximum length of 2,048. They are optimized and accelerated with Deepspeed Zero3 and FlashAttention2. We use the AdamW optimizer with a *Linear* learning rate of 2e-5. The training epoch is set to 1.

**Prompt Examples.** To guide the model towards generating executable Python code, we prepend the following prompt before each input:

```
Write Python code to solve the question.
```

We illustrate the few-shot prompts used in our experiments. The following shows the training-time few-shot prompt (MATH_PROMPT_FS) and the test-time prompt (MATH_PROMPT_FS_TEST). The test-time prompt only contains the first example of training-time prompt.

Listing 1: Few-shot prompt for training (MATH_PROMPT_FS)

```
The following are three examples for reference.

Example 1:
The question is : Olivia has $23. She bought five bagels for $3 each.
How much money does she have left?
The solution code is:
```python
def solution():
    '''Olivia has $23. She bought five bagels for $3 each.
    How much money does she have left?'''
    money_initial = 23
    bagels = 5
    bagel_cost = 3
    money_spent = bagels * bagel_cost
    money_left = money_initial - money_spent
    result = money_left
    return result
```

... (Examples 2 and 3 omitted for brevity)
```

## B  TEST TASKS AND BENCHMARK

Table 4 lists the benchmark tasks used in our experiments. Below we provide more detailed descriptions of each dataset: the types of math problems included, what makes them hard or easy, and an example from each.

### B.1  DATASET DESCRIPTIONS

- **GSM8K (Grade School Math 8K)** (Cobbe et al., 2021) This dataset contains approximately 8,500 linguistically diverse grade-school level word problems. Problems require between 2 to 8 reasoning steps and use basic arithmetic operations (addition, subtraction, multiplication, division). The problems are designed to be solvable without advanced mathematics, but test multi-step reasoning and managing intermediate fractional or decimal computations.

- **GSM-Hard** (Gao et al., 2023) A held-out or more challenging subset related to GSM8K, designed to test generalization under harder or out-of-distribution settings. It shares the same format but contains examples that are less similar to the training distribution.

- **SVAMP** (Patel et al., 2021) Consists of 1,000 math word problems constructed by applying perturbations to existing datasets (such as ASDiv), adding irrelevant information or changing problem structure to challenge robustness. Each problem typically has one unknown variable, with no more than two mathematical expressions.

- **ASDiv** (Miao et al., 2020) Contains 2,305 word problems spanning a variety of types, with greater lexical variety, more diverse wording, variable placements, and reasoning patterns. Problems vary from relatively simple to fairly complex, testing both arithmetic and reasoning about relationships.

### B.2  EXAMPLE INSTANCES

To illustrate the characteristics of different datasets, we present representative examples as follows:

- **GSM8K**
  *Q:* Janet's ducks lay 16 eggs per day. She eats 3 for breakfast and bakes with 4. She sells the remainder at the market for $2 per egg.
  *A:* 18

- **GSM-Hard**
  *Q:* A robe takes 2,287,720 bolts of blue fiber and half that much white fiber. How many bolts in total does it take?
  *A:* 3,431,580

- **SVAMP**
  *Q:* There are 87 oranges and 290 bananas. If the bananas are organized into 2 groups, how big is each group of bananas?
  *A:* 145

- **ASDiv**
  *Q:* Seven red apples and two green apples are in the basket. How many apples are in the basket?
  *A:* 9

Table 4: Benchmark tasks used in our experiments.

| Domains | Task name | Is Held-out? | Test Samples | Max Length | Sources |
|---|---|:---:|:---:|:---:|:---:|
| Math Reasoning | GSM8K | | 1,319 | 2,048 | Cobbe et al. (2021) |
| | GSM-Hard | ✓ | 1,319 | 2,048 | Gao et al. (2023) |
| | SVAMP | ✓ | 1,000 | 2,048 | Patel et al. (2021) |
| | AsDiv | ✓ | 2,305 | 2,048 | Miao et al. (2020) |

## C  COMPUTATION OF SIMILARITY SCORES

To evaluate the diversity of reasoning trajectories, we define a similarity score based on trajectory embeddings.

**Setup.** For each problem instance with at least 10 trajectories, we align datasets by intersecting their `origin_id` sets. Each trajectory is embedded using `Qwen/Qwen3-Embedding-0.6B`, as $f(\cdot)$. Queries $t^q$ are prefixed with a short instruction describing the task of retrieving logically equivalent trajectories, while candidate trajectories $t^d$ are encoded directly. The instruction for retrieving query is:

```
task = 'Given a reasoning trajectory in code form, identify and retrieve
    those strictly similar in logic and structure'
return f'Instruction: {task}\nThe given trajectory: {query}'
```

This instruction guides the model to focus on logical and structural consistency rather than surface-level textual overlap

**Pairwise Similarity.** Let $E \in \mathbb{R}^{n \times d}$ denote the embeddings of $n$ trajectories. We compute the cosine similarity matrix

$$S = E \cdot E^\top.$$

Self-similarities on the diagonal are masked out. The similarity score for an instance is then

$$\text{Sim}_{\text{instance}} = \frac{1}{n(n-1)} \sum_{i=1}^{n} \sum_{\substack{j=1 \\ j \neq i}}^{n} \langle e_i, e_j \rangle,$$

where $\langle e_i^q, e_j^d \rangle$ denotes cosine similarity between embeddings $e_i^q = f(t_i^q)$ and $e_j^d = f(t_j^d)$.

**Dataset-Level Score.** The dataset-level similarity is the mean over all valid instances:

$$\text{Sim}_{\text{dataset}} = \frac{1}{|\mathcal{D}|} \sum_{k \in \mathcal{D}} \text{Sim}_{\text{instance}}^{(k)}.$$

**Visualization.** We plot histograms of similarity scores across datasets and mark the median with dashed lines, enabling analysis of both central tendency and diversity, as shown in Figure 5. Lower similarity reflects richer trajectory diversity, while higher similarity indicates redundancy.

# D COMPUTATIONAL COST AND EFFICIENCY ANALYSIS

## D.1 MEASUREMENT PROTOCOL

We report the full computational cost of our self-evolution framework, including generation, verification, and SFT fine-tuning. All experiments are conducted on 4×RTX3090 GPUs (24GB each) using DeepSpeed ZeRO3 with FlashAttention2. Wall-clock time is measured from job start to completion, including I/O and synchronization. FLOPs are estimated following common practice(Kaplan et al., 2020; Hoffmann et al., 2022):

$$\text{FLOPS}_{\text{infer}} = f \times N_{\text{params}} \times N_{\text{infer-tokens}},$$

$$\text{FLOPS}_{\text{train}} = g \times N_{\text{params}} \times N_{\text{train-tokens}},$$

where $N_{\text{params}}$ is the model size(1.8B parameters for InternLM2.5-1.8B), $f$ and $g$ denote the average FLOPs-per-token multipliers for inference and training respectively. We empirically measured $f = 2$ and $g = 6$ on InternLM2.5-1.8B(KV-cache enabled).

## D.2 OVERALL COMPUTATIONAL COST

In our framework, the total computation mainly comes from three stages: *Trajectory Exploration* and *Trajectory Rethinking* during inference, and the subsequent SFT training after *Trajectory Selection*. Based on the following equations, we compute the corresponding numbers of inference tokens $N_{\text{infer-tokens}}$:

$$N_{\text{infer-tokens-explore}} = N_{\text{dataset}} \times I \times K \times (\bar{L}_{\text{Question}} + \bar{L}_{\text{Explore}}),$$

$$N_{\text{infer-tokens-rethink}} = N_{\text{dataset}} \times I \times K \times (\bar{L}_{\text{Question}} + \bar{L}_{\text{Rethink}} - \frac{\beta}{2} \times \bar{L}_{\text{Explore}}),$$

where $\bar{L}_{\text{Question}}$, $\bar{L}_{\text{Explore}}$, and $\bar{L}_{\text{Rethink}}$ denote the average token lengths of the question, exploration, and rethinking parts, respectively, and $\beta$ is the maximum truncation ratio for rethinking.

The total number of training tokens used in the subsequent SFT stage for each iteration is given by

$$N_{\text{train-tokens}} = N_{\text{selected}} \times (\bar{L}_{\text{Question}} + \bar{L}_{\text{Answer}}),$$

where $N_{\text{selected}}$ denotes the number of selected trajectories per input after *Trajectory Selection*, and $\bar{L}_{\text{Answer}}$ is the average token length of the selected answers.

For comparison with ENVISIONS, from the perspective of our formulation, its framework can also be decomposed into three stages: *Exploration*, *Refinement*, and *Training*. Among them, the inference token count of the *Refinement* stage can be defined as

$$N_{\text{infer-tokens-refine}} = N_{\text{dataset}} \times I \times K \times (\bar{L}_{\text{Question}} + \bar{L}_{\text{Refine}} + \bar{L}_{\text{Explore}}),$$

while the other two parts (*Exploration* and *Training*) can be analogously formulated following the equations above.The comprehensive results, including wall-clock time and other statistics, are summarized in 5.

Table 5: **Overall computational cost** of the self-evolution framework.

| Stage | #GPUs | Avg seq len (L) | FLOPs ($\times 10^{15}$) | Wall-clock (h) |
|---|---|---|---|---|
| **Ours**($I = 10, K = 5$) | | | | |
| Exploration | 4 | 130.1 | 475.0 | |
| Rethinking | 4 | 139.8 | 373.0 | 139.3 |
| SFT | 4 | 136.8 | 935.5 | |
| **Total** | – | – | **1783.5** | |
| **ENVISIONS**($I = 10, K = 5$) | | | | |
| Exploration | 4 | 119.9 | 450.7 | |
| Refinement | 4 | 123.7 | 751.9 | 152.0 |
| SFT | 4 | 122.3 | 856.6 | |
| **Total** | – | – | **2059.2** | |

### D.3 COST–PERFORMANCE TRADE-OFF

To assess the overall efficiency of the proposed self-evolution framework, we compare its computational overhead and performance gains against the ENVISIONS baseline. As summarized in Table 5, our method requires fewer overall FLOPs ($1.78 \times 10^{18}$ vs. $2.06 \times 10^{18}$) and achieves a slightly shorter wall-clock time per full iteration (139.3h vs. 152.0h).

A stage-wise analysis reveals the source of this improvement. Our framework incurs higher computational cost during both the *Exploration* (475.0 vs. 450.7) and *SFT* (935.5 vs. 856.6) stages. The efficiency gain instead arises primarily from the optimized intermediate stage: the *Trajectory Rethinking* cost (373.0) is substantially lower than the *Refinement* stage of ENVISIONS (751.9). This indicates that the overall cost reduction is not uniform across stages, but is driven by the more efficient rethinking procedure that eliminates redundant refinement steps while preserving trajectory quality.

To further evaluate cost-effectiveness, we normalize performance gains by computational cost relative to the few-shot baseline. Across all datasets and Pass@K metrics (Table 1), our approach consistently improves accuracy while maintaining competitive cost. For example, on GSM8K, our method improves Pass@16 from 63.53% to 71.80%, an absolute gain of 8.27%. Given a total cost of $1.78 \times 10^{18}$ FLOPs and 152.0 wall-clock hours, this corresponds to roughly **0.046% Pass@16 gain per $10^{15}$ FLOPs**, and **0.059% Pass@16 gain per wall-clock hour**.

Similar trends hold across remaining datasets, indicating that the proposed entropy-aware self-evolution framework achieves a more favorable cost–performance ratio than ENVISIONS. Overall, the results suggest that our design improves both computational efficiency and return on compute investment, particularly due to the substantially streamlined intermediate rethinking stage.

## E THEORETICAL JUSTIFICATION FOR ENTROPY-BASED EXPLORATION

### E.1 ENTROPY AS A PRACTICAL SURROGATE FOR EPISTEMIC UNCERTAINTY

While various uncertainty measures exist—such as mutual-information–based acquisition (Houlsby et al., 2011) or epistemic/aleatoric decomposition via Bayesian approximations (Kendall & Gal, 2017)—we adopt sequence-level Shannon entropy due to its computational simplicity and its direct alignment with the model's predictive distribution. Importantly, entropy admits a closed-form linkage to the expected supervised loss, implying that high-entropy trajectories contribute proportionally stronger gradient signals during fine-tuning. Although entropy alone does not separate epistemic from aleatoric uncertainty, our pipeline mitigates this limitation through a verifier and token-level rethinking stage that retains only trajectories both correct and uncertain. This filtering suppresses irreducible noise and allows entropy to function as an effective proxy for epistemic uncertainty in practice.

### E.2 Entropy as an Exploration-Enhancing Signal

Beyond its analytic connection to the expected supervised loss (Eq. (9)–(12)), entropy selection is theoretically grounded as a signal for exploration. We provide two complementary perspectives.

**Bayesian Active Learning** In Bayesian active learning, predictive entropy $H[y \mid x, \mathcal{D}]$ provides an upper bound on the mutual information between the model parameters $\theta$ and the labels $y$ (Houlsby et al., 2011):

$$I[y; \theta \mid x, \mathcal{D}] = H[y \mid x, \mathcal{D}] - \mathbb{E}_{\theta \sim p(\theta \mid \mathcal{D})}\big[H[y \mid x, \theta]\big] \leq H[y \mid x, \mathcal{D}]. \tag{13}$$

High-entropy samples therefore indicate high potential information gain, effectively targeting points that reduce epistemic uncertainty.

**Maximum-Entropy Reinforcement Learning** From a reinforcement learning perspective, maximum-entropy formulations encourage broader exploration and prevent premature convergence to overconfident modes (Ziebart, 2010; Haarnoja et al., 2018). The objective can be written as:

$$\pi^* = \arg\max_{\pi} \mathbb{E}_{\tau \sim \pi}\left[\sum_{t=0}^{T} r(s_t, a_t) + \alpha H\big(\pi(\cdot \mid s_t)\big)\right], \tag{14}$$

where $H(\pi(\cdot \mid s_t))$ is the policy entropy and $\alpha$ is a temperature parameter controlling exploration. Learning from verified high-entropy trajectories similarly encourages the model to expand its reasoning space beyond currently confident solutions.

### E.3 Integration into Our Pipeline

By combining entropy selection with a verification stage, our pipeline ensures that retained high-entropy trajectories are both informative and correct, effectively suppressing aleatoric noise while promoting structured exploration. This provides a principled justification for using entropy as a practical surrogate for epistemic uncertainty.

### E.4 Entropy, Expected Loss, and Mutual Information: A Formal Link

We formalize the connection between sequence-level Shannon entropy, expected supervised loss, and mutual information as follows:

**Entropy as a Surrogate for Expected Loss and Information Gain** Let $p_\theta(y \mid x)$ be the predictive distribution of a model parameterized by $\theta$. Then the expected supervised cross-entropy loss for a candidate sample $x$ is

$$\mathbb{E}_{y \sim p_\theta}[-\log p_\theta(y \mid x)] = H[y \mid x, \theta], \tag{15}$$

and the predictive entropy satisfies

$$H[y \mid x, \mathcal{D}] = \mathbb{E}_{\theta \sim p(\theta \mid \mathcal{D})}[H[y \mid x, \theta]] + I[y; \theta \mid x, \mathcal{D}], \tag{16}$$

where $I[y; \theta \mid x, \mathcal{D}]$ is the mutual information between $y$ and $\theta$ given data $\mathcal{D}$. Consequently, high predictive entropy $H[y \mid x, \mathcal{D}]$ implies both higher expected supervised loss and higher potential reduction in epistemic uncertainty.

**Proof** By definition, the expected supervised cross-entropy loss for a model sample $x$ is

$$\mathbb{E}_{y \sim p_\theta}[-\log p_\theta(y \mid x)] = H[y \mid x, \theta]. \tag{17}$$

Taking the expectation over the posterior $p(\theta \mid \mathcal{D})$, we have

$$\mathbb{E}_{\theta \sim p(\theta \mid \mathcal{D})}[H[y \mid x, \theta]]. \tag{18}$$

The predictive entropy decomposes as

$$H[y \mid x, \mathcal{D}] = I[y; \theta \mid x, \mathcal{D}] + \mathbb{E}_{\theta \sim p(\theta \mid \mathcal{D})}[H[y \mid x, \theta,] \tag{19}$$

which follows directly from the standard mutual information identity:

$$I[y; \theta \mid x, \mathcal{D}] = H[y \mid x, \mathcal{D}] - \mathbb{E}_\theta[H[y \mid x, \theta]]. \tag{20}$$

Therefore, a sample with higher predictive entropy contributes proportionally higher expected supervised loss and has higher mutual information, justifying its selection for exploration.

## E.5 THEORETICAL JUSTIFICATION FOR HIGH-ENTROPY TRUNCATION

In this section, we provide theoretical motivation for why truncating a trajectory at *high-entropy tokens* and re-sampling from these positions can effectively increase trajectory diversity and improve downstream reasoning performance.

**High-Entropy Tokens as Branching Points.** Let $p_t(\cdot)$ denote the model's token distribution at generation step $t$, and let $H_t = H(p_t)$ be its Shannon entropy. We define the *local branching factor* at position $t$ as

$$B_t \approx \exp(H_t). \tag{21}$$

When $H_t$ is small, $B_t \approx 1$ and the token distribution is almost deterministic, contributing little to the branching structure of the trajectory. In contrast, high-entropy positions ($H_t \gg 0$) correspond to *decision forks*: choices made at these tokens lead to divergent future trajectories. Under a multiplicative approximation of trajectory branching,

$$\#\text{Trajectories} \propto \prod_{t=1}^{T} B_t, \tag{22}$$

so a small set of high-entropy positions dominates the combinatorial expansion of reachable reasoning paths. Thus, re-sampling at high-entropy tokens is significantly more compute-efficient for increasing diversity than sampling uniformly across the sequence.

**Information-Theoretic View: Mutual Information Peaks.** Recent work has shown that during multi-step reasoning, some positions exhibit *mutual information peaks* with respect to the final answer. These positions—sometimes called "thinking tokens"—tend to be exactly the same high-entropy decision points where the model is most uncertain but also most informative. Formally, let $A$ denote the final answer and let $X_t$ be the token at step $t$. Information-theoretic analyses demonstrate that

$$I(X_t; A) \tag{23}$$

often exhibits sharp peaks at the same locations where $H_t$ is high. Perturbing or re-sampling at these positions thus explores distinct logical branches that meaningfully affect the correctness of the final answer. This observation aligns with recent studies on reasoning dynamics in LLMs(Qian et al., 2025), which empirically identify such MI peaks.

**High-Entropy Minority Tokens Drive Major Reasoning Variance.** Empirical analyses further suggest that a small fraction of tokens with the highest entropy account for the majority of reasoning variance. Specifically, the "high-entropy minority tokens" framework(Wang et al., 2025a) demonstrates that: (i) the distribution of token entropies in chain-of-thought reasoning is heavy-tailed, and (ii) the top 15–20% of tokens (ranked by entropy) correspond to the critical branching points that drive most of the performance variation in reinforcement learning or self-improvement updates. This theory directly supports our decision to truncate at high-entropy tokens and re-sample from these fork points.

**Connection to Gradient Efficiency.** From an optimization perspective, high-entropy tokens also correspond to positions with the largest variance in the model's predictive distribution. Updating or re-sampling at these locations yields the greatest marginal benefit, whereas modifying low-entropy (near-deterministic) positions provides negligible gains. This reinforces the rationale that high-entropy truncation is a principled and compute-efficient mechanism for exploring alternative reasoning paths.

Together, the multiplicative branching model, mutual-information analysis, and high-entropy minority token theory provide a coherent justification: *high-entropy tokens serve as the key decision points in a reasoning trajectory*; therefore, truncating and re-sampling at these positions maximizes trajectory diversity per unit compute and improves the probability of discovering correct reasoning paths.

## F    COMPARISON WITH ENVISIONS

Both ENVISIONS and our framework leverage external validators to select positive samples based on reliable feedback, which are then used for self-training. The key differences are as follows:

**Trajectory Generation Strategy**: ENVISIONS employs a self-refine mechanism, where the model uses previously generated trajectories as a basis to revise and regenerate them. In contrast, our framework uses a rethinking mechanism, where the model continues generating new trajectories from high-entropy truncations.

**Positive Sample Selection**: ENVISIONS relies on a self-reward mechanism, selecting high-confidence samples as training positives. Our framework adopts an entropy-aware selection strategy, prioritizing high-entropy trajectories.

Both frameworks follow a similar explore–refine–selection pipeline with a validator, which is why we include ENVISIONS as a baseline in our experiments.

## G    TRAJECTORY EXAMPLES

Tables 7 and 8 present several example generation trajectories under self-evolution across multiple iterations. From these observations, it can be seen that our method can occasionally find the correct solution more quickly when handling moderately difficult problems. For instance, as shown in Table 7, both methods produce identical solutions at iteration 2 for a given problem, yet our method discovers the correct solution already by iteration 4.

Moreover, after reviewing several representative samples, we observe that under our method, the model tends to leverage more annotated reasoning steps rather than relying solely on code. Across iterative rounds, our method also explores more diverse trajectories. In contrast, ENVISIONS tends to converge to similar trajectories once the correct solution is found; for example, in Table 8, the responses at iterations 8 and 10 are nearly identical.

## H    HYPERPARAMETER ANALYSIS

To investigate the effect of different hyperparameters in our framework, we conduct controlled experiments using InternLM2.5-1.8B trained on a small subset of 1,000 samples and additionally provide theoretical analysis for several key hyperparameter choices.

### H.1    ANALYSIS OF SAMPLING TEMPERATURE ON EVALUATION

To evaluate the influence of sampling temperature, we test the trained model on the GSM8K test set using sampling temperatures ranging from 0.6 to 1.2. The results in Figure 7 show that performance increases as temperature rises and subsequently decreases at higher temperatures, indicating that sampling temperature indeed affects output diversity and thus impacts Pass@K accuracy. Importantly, our method consistently outperforms ENVISIONS across wide range of tested temperatures, suggesting that the improvements are not merely a consequence of temperature effects but stem from the proposed self-evolution mechanism.

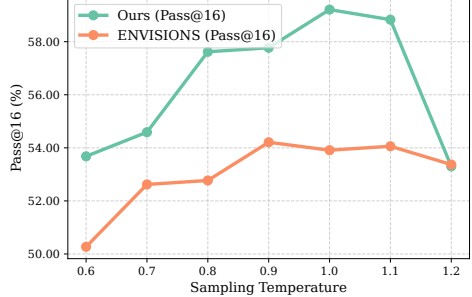

Figure 7: Pass@16 accuracy on the GSM8K test set under different sampling temperatures (0.6–1.2) for InternLM2.5-1.8B trained on a 1k-sample subset.

## H.2 ANALYSIS OF LOW-BUDGET PERFORMANCE AND SAMPLE EFFICIENCY

To evaluate whether the improvements arise solely from wider sampling at large $K$, rather than reflecting better sample efficiency, we further assess the model in the low-budget regime. Using InternLM2.5-1.8B trained on a subset of 1,000 samples, we report Pass@1, Pass@2, Pass@4, and Pass@8 on GSM8K. As shown in Table 6, our method consistently outperforms ENVISIONS even at small $K$, indicating that the gains are not restricted to large-batch exploration but also enhance single-shot and low-sample reasoning performance.

Table 6: Pass@K accuracy on GSM8K for InternLM2.5-1.8B trained on a 1k-sample subset.

| K | 1 | 2 | 4 | 8 |
|---|---|---|---|---|
| **Ours** | 29.34 | 33.74 | 41.24 | 46.93 |
| **ENVISIONS** | 26.16 | 30.48 | 38.13 | 44.66 |

## H.3 ANALYSIS OF TRUNCATION PARAMETERS

**Fraction of Top-entropy Tokens $\alpha$.** The parameter $\alpha$ controls which high-entropy tokens are considered as candidate truncation points. Prior work (Wang et al., 2025a) shows that reasoning trajectories contain a heavy-tailed entropy distribution in which roughly the top 15–20% of tokens contribute most to branching and downstream performance ("high-entropy minority tokens"). Setting $\alpha = 20\%$ therefore concentrates rethinking on the key decision forks while excluding low-entropy or weakly informative positions. Based on these theoretical insights, we recommend choosing $\alpha$ within the range $[0.15, 0.25]$.

**Maximum Truncation Ratio $\beta$.** The parameter $\beta$ determines the proportion of the original trajectory that is retained before applying high-entropy truncation and regeneration. If $\beta$ is set too small, the truncation point will lie excessively early in the reasoning process, making it unlikely to cover the high-entropy decision forks that drive trajectory diversity. In such cases, the model cannot effectively leverage the useful intermediate reasoning already present in the original trajectory. Conversely, if $\beta$ is set too large, the truncation occurs too late, leaving little room for regeneration and thereby limiting the diversity of alternative reasoning paths.

Balancing these two factors, we adopt $\beta = 0.8$, which retains sufficient prefix context to preserve meaningful reasoning structure while still allowing regeneration to explore new branches around high-entropy positions. As a general guideline, $\beta$ should be chosen to keep the truncation point within the region where high-entropy tokens typically occur; in practice, values in the range $[0.7, 0.85]$ provide a reasonable trade-off between leveraging existing reasoning and maintaining diversity.

## H.4 ANALYSIS OF SELF-EVOLUTION PARAMETERS

For the hyperparameters: $K$, $N$, $I$, we followed the same setup as Xu et al. (2025) to ensure a fair comparison and did not conduct additional experiments to explore their parameter choices.

For reproducibility, the sampling budget $K$ controls the number of trajectories generated per input during exploration: too small $K$ reduces coverage of useful reasoning paths, while too large $K$ (e.g., $K = 10$ or 15) can improve performance on some specific tasks but does not generalize consistently (Xu et al., 2025).

The selection size $N$ specifies how many trajectories per iteration are used for SFT: larger $N$ increases training diversity, but excessively large $N$ may introduce noise and reduce the selectivity of the dataset; thus, $N$ should be chosen in conjunction with $K$.

Finally, the iteration count $I$ determines how many self-evolution cycles are performed; early iterations produce substantial gains by correcting systematic errors, while later iterations provide incremental refinements. In our setup, the chosen $I$ is sufficient for the full dataset to reach a stable set of training trajectories in the final iterations, with the performance gains gradually diminishing.

In addition, we set the sampling temperature to 1 to capture the model's intrinsic behavior during trajectory generation, allowing the self-evolution process to explore diverse reasoning paths. A top-$p$ value of $0.95$ filters out extremely low-probability tokens, improving the efficiency and correctness of math reasoning by focusing on more plausible steps. Together, these settings ensure that sampled trajectories are both representative and reliable for self-finetuning.

| Method, Iterations, Correctness | Trajectories of Question: "How many even divisors does 7! have?" |
|---|---|
| Ours, iter. 2, wrong | ```python
def solution():
    '''How many even divisors does $7!$ have?'''
    result = 0
    for i in range(1, 7+1):
        if 7**(i) % i == 0 and 7**i % i == 0:
            result += 2
    return result
``` |
| ENVISIONS, iter. 2, wrong | ```python
def solution():
    '''How many even divisors does $7!$ have?'''
    result = 0
    for i in range(1, 7+1):
        if 7**(i) % i == 0 and 7**i % i == 0:
            result += 2
    return result
``` |
| Ours, iter. 4, correct | ```python
def solution():
    '''How many even divisors does $7!$ have?'''
    import math
    ''' $7! = 7 * 6 * 5 * 4 * 3 * 2 * 1$'''
    divisors = math.factorial(7)
    even_divisors = 0
    for i in range(1, divisors + 1):
        if divisors % i == 0 and i % 2 == 0:
            even_divisors += 1
    return even_divisors
``` |
| ENVISIONS, iter. 4, wrong | ```python
def solution():
    '''How many even divisors does $7!$ have?'''
    import math   # importing the math module
    result = math.factorial(7) / 2
    return result\n
``` |
| Ours, iter. 6, correct | ```python
def solution():
    '''Solution: How many even divisors does $7!$ have?'''
    import math
    divisors = []
    result = math.factorial(7)
    for i in range(1, result+1):
        if result % i == 0:
            if i % 2 == 0:
                divisors.append(i)
    return len(divisors)
``` |
| ENVISIONS, iter. 6, wrong | ```python
def solution():
    '''How many even divisors does $7!$ have?'''
    evens_initial = 0
    from math import factorial, factorial
    for i in range(1, 7+1):
        divisors = factorial(i)
            if divisors % 2 == 0:
                evens_initial += 1
    result = evens_initial
    return result
``` |

Table 7: Generated trajectory examples for iteration 2, 4 and 6.

| Method, Iterations, Correctness | Trajectories of Question: "How many even divisors does 7! have?" |
|---|---|
| Ours, iter. 8, correct | ```python
def solution():
    '''How many even divisors does $7!$ have?'''
    import math
    # Convert factorial to a string as factorial is not
    # multiplicative in numbertheory.
    # Here we are using the formula
    # factorial(n) = n * (n-1) * (n-2) *...2 * 1
    # factorial(7) = 7! = 7 * 6 * 5 * 4 * 3 * 2 * 1
    # digital product 10
    hommiealpha = math.factorial(7)
    divisors = []
    for i in range(1, hommiealpha+ 1):
        if 7 * 6 * 5 * 4 * 3 * 2 * 1 % i == 0:
        if i % 2 == 0:
        divisors.append(i)    #OD
    return len(divisors)
``` |
| ENVISIONS, iter. 8, correct | ```python
def solution():
    '''How many even divisors does $7!$ have?'''
    import math
    result = 0
    for i in range(1, 10000):
        if math.factorial(7) % i == 0:
            if i % 2 == 0:
                result += 1
    return result
``` |
| Ours, iter. 10, correct | ```python
def solution():
    '''How many even divisors does $7!$ have?'''
    from math import factorial
    result = factorial(7)
    # import it as factorial function from math module
    # using module to get all the even divisors found
    return len([i for i in range(1, result + 1)
                    if result % i == 0 and i % 2 == 0])
``` |
| ENVISIONS, iter. 10, correct | ```python
def solution():
    '''How many even divisors does $7!$ have?'''
    import math
    even_divisors = 0
    for i in range(1, 10000):
        if math.factorial(7) % i == 0:
            if i % 2 == 0:
                even_divisors += 1
    result = even_divisors
    return result
``` |

Table 8: Generated trajectory examples for iteration 8 and 10.

