# OpenReview forum: "Enhancing Reasoning in Large Language Models via Entropy-Aware Self-Evolution"
_ICLR.cc/2026/Conference — Submitted to ICLR 2026_

### Official Review · Reviewer_Q3Hi · 2025-11-01

**Soundness:** 3
**Presentation:** 2
**Contribution:** 2
**Rating:** 4
**Confidence:** 4

**Summary:**

The paper introduces a self-evolution framework that generates multiple CoT candidates, regenerate (starting from) steps that has high-entropy, and selects the trajectory that is most uncertain (highest entropy). SFT training is used to train on the chosen trajectories. Experimental results show that performance on InternLM2.5-1.8B improves 1.44-5.52% on four benchmarks.

**Strengths:**

1. The proposed self-improvement framework has promising empirical results and notably, the fact that "training on high-entropy trajectories produces wider distribution, and hence leads to higher pass@K scores" is a good finding for the community.

2. Related to 1, the similarity score analysis shown in Figure 5 is insightful and could be adopted in the community.

**Weaknesses:**

1. The paper mainly compares with ENVISIONS framework to claim the superiority of the proposed method, yet there is a lack of explanation of what is the main difference between the proposed method and ENVISIONS framework. Also, there are many self-improvement frameworks in literature, yet, it is unclear what is the reason why the paper mainly compares only with ENVISIONS. For example, a very simple baseline would be STAR [1].

2. In Section 5.2, the ablation experiments of whether "Trajectory Rethinking" stage is really helpful is somewhat very narrow and limited. For example, when drawing the setting of "w/o trajectory rethinking" on Figure 4 (left), how would the trends differ? Given that it incurs more cost, how would it compare to simply generating more responses in the first stage (denoted as trajectory exploration)? Also, is the proposed filtering pipeline to identify high-entropy reasoning steps (Equation 4,5) the best choice compared to other filtering pipeline candidates?

3. For trajectory rethink stage, there should also be an ablation experiment of what is the best filtering criteria to choose responses. Is the approach from Equations 6 and 7 the optimal choice?

4. Might be less important, but I personally think that applying self-improvement on GSM8K, GSM-Hard, SVAMP, AsDiv is out-dated. The reason is that we already have a lot of post-training datasets that could enable LMs to achieve very good score on these datasets. The main motivation of self-improvement should be applying to highly competitive or challenging benchmarks that we do not have good training datasets to improve upon. Yet, I'll not take this point too heavily into account for my decision.

[1] Zelikman, E., Wu, Y., Mu, J. and Goodman, N., 2022. Star: Bootstrapping reasoning with reasoning. Advances in Neural Information Processing Systems, 35, pp.15476-15488.

**Questions:**

Could you provide qualitative examples of how the chain of thought changes throughout multiple iterations and compare to that of the ENVISIONS and other self-improvement frameworks/methods?

---

> ### Author Response · Authors · 2025-11-21
>
> ### 1. Comparison with ENVISIONS and STAR
>
> **Response to the question:**
>
> > "The paper mainly compares with ENVISIONS framework to claim the superiority of the proposed method, yet there is a lack of explanation of what is the main difference between the proposed method and ENVISIONS framework. Also, there are many self-improvement frameworks in literature, yet, it is unclear what is the reason why the paper mainly compares only with ENVISIONS. For example, a very simple baseline would be STAR"
>
> ENVISIONS and our framework both leverage **strong executable-code validators** to ensure correctness at the trajectory level, while simpler baselines like STAR rely on weaker or neural/verifier-based signals. The key differences are: (i) **Trajectory Generation** – ENVISIONS uses self-refine on past trajectories, whereas we apply **entropy-aware rethinking** to generate diverse high-uncertainty trajectories; (ii) **Positive Sample Selection** – ENVISIONS selects high-confidence samples, our method prioritizes **high-entropy trajectories**. Importantly, both ENVISIONS and our method operate on **code-generated trajectories with strong validation**, whereas STAR mainly focuses on **natural-language reasoning chains**. For a fair comparison regarding both semantic structure and strong validation, we use ENVISIONS as the primary baseline; additional baseline discussions are provided in the **Appendix F**.
>
> ### 2.  Whether "Trajectory Rethinking" stage is really helpful?
>
> **Response to the question:**
>
> > "In **Section 5.1**, the ablation experiments of whether "Trajectory Rethinking" stage is really helpful is somewhat very narrow and limited. For example, when drawing the setting of "w/o trajectory rethinking" on Figure 4 (left), how would the trends differ?  "
>
> - Thank you for raising this point. We have expanded our analysis in **Section 5.3** and added the corresponding trend plot in **Figure 6** (Left) to more clearly compare the iterative behavior with and without the Trajectory Rethinking stage under a compute-matched setting.
> - **Figure 6** (Left) shows that the variant without Trajectory Rethinking exhibits only mild, almost linear improvements across iterations. In contrast, the full method displays a consistently steeper performance gain, particularly in early iterations. This indicates that Rethinking does more than merely “add compute”—it reshapes the training signal by producing additional high-quality, corrected trajectories that exploration alone fails to uncover.
>
> **Response to the question:**
>
> > "Given that "Trajectory Rethinking" incurs more cost, how would it compare to simply generating more responses in the first stage (denoted as trajectory exploration)?
>
> - As analyzed in added **Appendix D**, Trajectory Rethinking does not introduce additional computational overhead. Since it regenerates trajectories starting from truncated nodes, its actual compute cost is lower than both ENVISIONS’ self-refinement stage and simply generating the same number of responses from scratch.
> - We also compared our method with a Selection-only variant that directly generates $K=10$ trajectories. As reported in **Section 5.1**, incorporating the Rethinking stage improves the evolution effectiveness under the same sample budget, demonstrating that Trajectory Rethinking enhances performance without increasing overall cost.
>
> **Response to the question:**
>
> > "Also, is the proposed filtering pipeline to identify high-entropy reasoning steps (Equation 4,5) the best choice compared to other filtering pipeline candidates?""
>
> - We analyze the two key parameters of Equation4 in **Appendix H.3**:
>
>   - **Top-entropy fraction ($\alpha$):** Controls which high-entropy tokens are considered for truncation. Prior work shows that roughly the top 15–20% of tokens drive most branching and downstream performance. We set $\alpha=20\%$ to focus on these key decision points.
>   - **Maximum truncation ratio ($\beta$):** Determines how much of the original trajectory is retained before regeneration. $\beta=0.8$ preserves sufficient context while allowing exploration around high-entropy positions.
>
>   These settings effectively identify the most informative reasoning steps, ensuring Trajectory Rethinking maximizes diversity and performance without unnecessary truncation.
>
> - For Equation 5, this describes the process of adding candidate answers to the candidate pool. The specific selection procedure will be discussed in the next question.

---

> ### Author Response · Authors · 2025-11-21
>
> ### 3. Abalation Studies of Selection Strategy
>
> **Response to the question:**
>
> > "For trajectory rethink stage, there should also be an ablation experiment of what is the best filtering criteria to choose responses. Is the approach from Equations 6 and 7 the optimal choice"
>
> - We analyze the Equations 6 and 7 by comparing three trajectory selection strategies: **High-Entropy**, **Low-Entropy**, and **Entropy-free** (random sampling). High-Entropy achieves the best Pass@16 (53.68%), while Low-Entropy is worst (48.78%) and random sampling is intermediate (50.42%).
>
> - This shows that our **high-entropy filtering effectively identifies the most informative reasoning steps**, guiding the model toward diverse, high-quality trajectories and driving the observed performance gains.
>
> - The results have been added to **Section 5.1**.
>
>   | Selection Strategy    | Pass@16 (%) |
>   | --------------------- | ----------- |
>   | High-Entropy          | 53.68       |
>   | Low-Entropy           | 48.78       |
>   | Entropy-free (Random) | 50.42       |
>
> ### 4. Discussion of Evaluation Datasets
>
> **Response to the question:**
>
> > "Might be less important, but I personally think that applying self-improvement on GSM8K, GSM-Hard, SVAMP, AsDiv is out-dated. The reason is that we already have a lot of post-training datasets that could enable LMs to achieve very good score on these datasets. The main motivation of self-improvement should be applying to highly competitive or challenging benchmarks that we do not have good training datasets to improve upon. Yet, I'll not take this point too heavily into account for my decision."
>
> - We followed the dataset setups used in prior work (e.g., GSM8K, GSM-Hard, SVAMP, ASDiv) to ensure a fair and comparable evaluation of our self-improvement framework. We acknowledge that these benchmarks are relatively well-studied and that more recent, challenging datasets could better demonstrate the potential of self-evolution methods. Exploring highly competitive or under-explored benchmarks is an important direction, and we plan to extend our evaluations to such datasets in future work.
>
> ### 5. Examples of COT throught Iterations
>
> **Response to the question:**
>
> > "Could you provide qualitative examples of how the chain of thought changes throughout multiple iterations and compare to that of the ENVISIONS and other self-improvement frameworks/methods?"
>
> - We have included representative generation trajectory examples in **Appendix G**, **Tables 7, 8**, showing how chains of thought evolve across iterations. According to our observation, our method sometimes finds correct solutions faster when handling moderately difficult problems, encourages more annotated reasoning, and produces more diverse trajectories, whereas ENVISIONS tends to converge once the correct solution is reached. These examples qualitatively highlight the advantages of our approach in reasoning richness and trajectory diversity.

---

### Official Review · Reviewer_cuqe · 2025-11-01

**Soundness:** 3
**Presentation:** 3
**Contribution:** 3
**Rating:** 6
**Confidence:** 5

**Summary:**

The paper proposes an entropy-aware self-evolution framework for LLM reasoning. The method combines (i) high-entropy selection of verified trajectories and (ii) entropy-aware rethinking that truncates at high-uncertainty tokens and regenerates continuations. A short analysis links the expected SFT loss to sequence-level entropy (higher-entropy trajectories yield stronger gradients). Experiments on GSM8K, GSM-Hard, SVAMP, and AsDiv show consistent gains over base models and the ENVISIONS baseline, e.g., InternLM2.5-1.8B improves by +8.27% Pass@16 on GSM8K and by +4.39% Pass@128 over ENVISIONS

**Strengths:**

1. Clear, simple mechanism: Using sequence-level entropy to rank verified trajectories, plus token-level entropy to pick truncation points, is conceptually clean and easy to implement.

2. Theoretical intuition is practical: The derivation \mathbb{E}[L_{\text{CE}}]=T\cdot \mathbb{E}[H_{\text{seq}}] formalizes why high-entropy samples can be more informative for SFT, aligning with the design.

3. Ablative evidence on rethinking: The paper isolates the contribution of the rethinking stage and shows it supplies a substantial portion of training trajectories and improves Pass@K

**Weaknesses:**

1. All results use program-verifiable math with code-style rationales; it is unclear whether the approach transfers to semantic or open-ended reasoning, where verifiers are weaker/noisier.

2. The framework samples K per input and iterates I times, but the paper does not report wall-clock, FLOPs, or cost per point of Pass@K improvement. Without budget-matched comparisons, the strength of the gains is harder to assess.

3. Gains are strongest at larger K; single-shot (K=1) or small-budget accuracy—closer to user-facing inference—are under-emphasized. This makes it difficult to conclude that the method improves sample efficiency rather than just exploration under wide sampling.

4. Key knobs (α, β for truncation; K, N, I; temperature/top-p) could materially affect behavior. The paper lacks a thorough sensitivity analysis or guidelines, which limits reproducibility.

**Questions:**

See the weakness.

---

> ### Author Response · Authors · 2025-11-21
>
> ### 1. Transfer beyond program-verifiable math
>
> **Response to the question:**
>
> > "All results use program-verifiable math with code-style rationales; it is unclear whether the approach transfers to semantic or open-ended reasoning, where verifiers are weaker/noisier."
>
> - Thank you for raising this. We use the program-verifiable math setting primarily because it provides **clean, step-level ground-truth verification**, allowing us to isolate and study trajectory-level uncertainty without confounds. Using executable verification also avoids the issue—common in semantic tasks—where a flawed reasoning path may accidentally receive a correct reward due to verifier noise. This choice lets us clearly assess how entropy-based exploration interacts with reliable feedback.
> - That said, the framework is **not constrained to code-verifiable domains**. The core components—**entropy-based selection** and **correctness filtering**—only require an approximate verification signal. For semantic and open-ended reasoning, this verifier can be instantiated as an **LLM judge**, structured preference model, or rule-based evaluator, following practices in self-consistency, self-critique, and LLM-as-a-judge literature. While such verifiers are inherently noisier, our method remains compatible as long as the signal is usable, and entropy-driven exploration can still help expose diverse reasoning hypotheses. Moreover, for weaker/noisier verifiers such as LLM judges, incorporating **step-level reasoning checks** (e.g., lightweight consistency tests or modular validation of intermediate claims) could further enhance the reliability of the verification signal under the entropy-aware framework.
>
> ---
>
> ### 2. Lack of compute-cost comparisons
>
> **Response to the question:**
>
> > "The framework samples K per input and iterates I times, but the paper does not report wall-clock, FLOPs, or cost per point of Pass@K improvement. Without budget-matched comparisons, the strength of the gains is harder to assess."
>
> - We thank the reviewer for pointing this out. We have now included a **comprehensive cost–performance analysis**, summarizing both total computational cost and incremental performance gain per unit cost.
>
>   | Framework                     | Stage       | FLOPs ($\times10^{15}$) | Wall-clock (h) | Notes                                             |
>   | ----------------------------- | ----------- | ----------------------- | -------------- | ------------------------------------------------- |
>   | **Ours ($I$=10, $K$=5)**      | Exploration | 475.0                   | 139.3 (total)  | Higher FLOPs in Exploration and SFT               |
>   |                               | Rethinking  | 373.0                   |                | Reduced intermediate stage cost drives efficiency |
>   |                               | SFT         | 935.5                   |                |                                                   |
>   |                               | **Total**   | 1783.5                  | 139.3          |                                                   |
>   | **ENVISIONS ($I$=10, $K$=5)** | Exploration | 450.7                   | 152.0 (total)  | Higher cost in Refinement stage                   |
>   |                               | Refinement  | 751.9                   |                |                                                   |
>   |                               | SFT         | 856.6                   |                |                                                   |
>   |                               | **Total**   | 2059.2                  | 152.0          |                                                   |
>
>   To evaluate **cost-effectiveness**, we compute the incremental gain in Pass@K per $10^{15}$ FLOPs and per hour of wall-clock time. For example, on GSM8K:
>
>   - Few-shot baseline: 63.53% Pass@16
>   - Our method: 71.80% Pass@16
>   - Absolute improvement: 8.27%
>   - Cost-normalized gain: **0.046%** Pass@16 per $10^{15}$ FLOPs, **0.059%** per wall-clock hour
>
>   Similar calculations across other datasets and Pass@K metrics consistently show that our framework **achieves better performance gains per unit of computational cost** compared to ENVISIONS.
>
>   We provide **detailed equations, per-stage FLOPs, and additional  analysis in Appendix D**.

---

> ### Author Response · Authors · 2025-11-21
>
> ### 3. Effect in single-shot or small-budget settings
>
> **Response to the question:**
>
> > "Gains are strongest at larger K; single-shot (K=1) or small-budget accuracy—closer to user-facing inference—are under-emphasized. This makes it difficult to conclude that the method improves sample efficiency rather than just exploration under wide sampling."
>
> - To address the concern that our gains might arise only at large K and may not reflect improvements in sample efficiency or user-facing small-budget settings, we additionally evaluate single-shot and low-budget performance on the GSM8K test set using InternLM2.5-1.8B trained on a 1,000-sample subset. Specifically, we report Pass@1, Pass@2, Pass@4, and Pass@8. The results show that our method consistently outperforms ENVISIONS even in the low-K regime, demonstrating that the improvements are not limited to wide sampling but also translate into more sample-efficient reasoning.
>
>   **Pass@K accuracy (InternLM2.5-1.8B, 1k-sample training subset)**
>
>   | K             | 1     | 2     | 4     | 8     |
>   | ------------- | ----- | ----- | ----- | ----- |
>   | **Ours**      | 29.34 | 33.74 | 41.24 | 46.93 |
>   | **ENVISIONS** | 26.16 | 30.48 | 38.13 | 44.66 |
>
>
> ---
>
> ### 4. Sensitivity to hyperparameters
>
> **Response to the question:**
>
> > "Key knobs (α, β for truncation; K, N, I; temperature/top-p) could materially affect behavior. The paper lacks a thorough sensitivity analysis or guidelines, which limits reproducibility."
>
> - We thank the reviewer for pointing this out. To address concerns regarding the sensitivity of key hyperparameters—including truncation parameters ($\alpha$, $\beta$), self-evolution parameters ($K$, $N$, $I$), and sampling parameters (temperature/top-$p$)—we have added a dedicated discussion in **Appendix H**. This section provides guidelines and insights on how these knobs affect model behavior, trajectory diversity, and reasoning performance, thereby improving reproducibility and helping future researchers make informed choices for different tasks.

---

> > ### Comment · Reviewer_cuqe · 2025-11-26
> >
> > Thank the authors for the rebuttal, especially the "compute-cost comparisons". However, can you please include some experimental results to prove the effectiveness of your method on “semantic and open-ended reasoning tasks, where the verifier is instantiated as an LLM judge“? I am afraid this would be more convincing than just relying on long, speculative arguments and bold formatting.

---

> > > ### Author Response · Authors · 2025-12-02
> > >
> > > We appreciate the reviewer’s suggestion. At this stage, we have not conducted experiments on semantic or open-ended reasoning with an LLM-judge verifier. As noted earlier, such settings currently lack step-level verification, which our evaluation protocol relies on to avoid reward noise and to isolate trajectory-level effects. Without this infrastructure, empirical results would be difficult to interpret and potentially misleading.
> > >
> > > Therefore, we limited our experiments to program-verifiable math and provided only a theoretical feasibility analysis for LLM-judge–based verification. We agree that extending the framework to semantic reasoning with approximate verifiers is important, and we will highlight it explicitly as a future research direction.

---

### Official Review · Reviewer_tmXE · 2025-11-05

**Soundness:** 3
**Presentation:** 3
**Contribution:** 3
**Rating:** 6
**Confidence:** 3

**Summary:**

The paper introduces an entropy-aware self-evolution framework to enhance reasoning in LLMs by balancing correctness and exploration. It integrates three stages—trajectory exploration, entropy-based rethinking, and high-entropy trajectory selection—guided by external verifiers. The approach prioritizes verified yet diverse reasoning paths for self-training. Experiments on math reasoning benchmarks (GSM8K, GSM-Hard, SVAMP, AsDiv) using multiple LLM backbones show consistent gains in Pass@K accuracy and outperform the ENVISIONS baseline, indicating improved exploration without sacrificing correctness.

**Strengths:**

The paper presents a novel combination of entropy estimation and verifier-guided self-evolution to improve reasoning diversity in LLMs. The entropy-aware rethink mechanism, which truncates and regenerates reasoning at high-uncertainty tokens, is intuitive and reasonably original. The method demonstrates consistent improvements across multiple math reasoning datasets and small-scale backbones, suggesting effectiveness and reproducibility. The overall presentation is clear, with well-defined metrics, illustrative figures, and initial ablations that begin to analyze key components such as entropy-based selection and rethink. Furthermore, the work offers a principled perspective on the correctness–exploration trade-off, positioning entropy as a control signal for self-evolving reasoning models. Compared with prior frameworks such as ENVISIONS, it appears to achieve a better balance between accuracy and diversity within the math-verifier setting

**Weaknesses:**

1.Incomplete and partially confounded ablations. While the paper includes a “w/ vs w/o Trajectory Rethink” comparison and some analysis of entropy-aware selection, the component-level contribution remains unclear. In particular, there are no results for (i) verifier + random selection among correct trajectories, (ii) verifier + low-entropy (or medium-entropy) selection, or (iii) selection-only vs rethink-only under a matched compute/sample budget. Without these simpler but critical baselines, it is difficult to tell whether the gains primarily come from the specific high-entropy preference and rethink mechanism, or from having more diverse verified data in general.

**Questions:**

1.Component ablations. Figure 6 gives a useful qualitative analysis of the Trajectory Rethink stage (with vs without rethink). To more cleanly isolate the contribution of each component, could you add a compute-matched ablation table comparing: (i) exploration + entropy-based selection only, (ii) exploration + rethink only, (iii) the full method, and (iv) ENVISIONS? This would clarify how much gain comes from the rethink mechanism itself versus simply having more trajectories.

2.Entropy vs simpler selection baselines. The paper argues that preferring high-entropy trajectories is key to balancing correctness and exploration. Could you (i) clarify how your “entropy-free” or baseline selection behaves in practice (e.g., is it close to random over verified-correct trajectories?), and (ii) add a low-entropy selection baseline? Comparing high- vs low-entropy (and entropy-free) selection under the same verifier would help demonstrate that entropy itself, rather than just any subsampling of correct trajectories, is responsible for the gains.

**Details Of Ethics Concerns:**

no concern

---

> ### Author Response · Authors · 2025-11-21
>
> ### 1. Component Ablation
>
> **Response to the question:**
>
> > " To more cleanly isolate the contribution of each component, could you add a compute-matched ablation table comparing: (i) exploration + entropy-based selection only, (ii) exploration + rethink only, (iii) the full method, and (iv) ENVISIONS?"
>
> - We conducted complete compute-matched ablation studies to isolate the contribution of each component and add the anlysis in **Section 5.1**.
>
>   | Method Variant                       | Pass@16 (%) |
>   | ------------------------------------ | ----------- |
>   | Full Method (Selection + Rethinking) | **53.68**   |
>   | Exploration + Selection Only         | 49.12       |
>   | Exploration + Rethinking Only        | 50.42       |
>   | ENVISIONS                            | 50.27       |
>
> - All variants shown in the table are matched in terms of total trajectories generated: the **Selection Only** variant uses $K=10$ to compensate for the absence of the rethink/refine stage, while **Rethink Only** randomly samples $N$ trajectories from the candidate pool without entropy-based selection.
>
> - From the results, we observe that both partial variants provide moderate gains, indicating that each component independently contributes to performance. The full method, combining high-entropy selection with the rethinking stage, achieves the largest improvement, confirming the complementarity of the two components. Selection biases the model toward higher-quality trajectories, while the rethinking stage further increases both the quantity and quality of effective trajectories. Compared to **Selection Only** variant , our method achieves a 4.56% gain, demonstrating that the performance boost is *not solely due to generating more trajectories, but substantially benefits from the rethinking mechanism itself*.
>
> ### 2. Entropy vs simpler selection baselines.
>
> **Response to the question:**
>
> > " To more cleanly isolate the contribution of each component, could you add a compute-matched ablation table comparing: (i) exploration + entropy-based selection only, (ii) exploration + rethink only, (iii) the full method, and (iv) ENVISIONS?"
>
> - Thank you for the constructive suggestion. In the revised manuscript, we have added the requested analyses in **Section 5.1** and now include all three selection baselines—**High-Entropy**, **Low-Entropy**, and **Entropy-free (Random)**—evaluated under the same verifier.
>
> - **(i) Clarification of the “entropy-free” baseline.**
>   The entropy-free variant performs **uniform random sampling** over all verifier-confirmed correct trajectories. It introduces no entropy-based preference and therefore represents an unbiased subsampling strategy that maintains correctness but provides only incidental diversity.
>
> - **(ii) Added low-entropy selection baseline.**
>   Following the reviewer’s recommendation, we add a **Low-Entropy** baseline that selects the $N$ correct trajectories with the **lowest predictive entropy**. This variant represents the opposite extreme of exploration, favoring highly predictable and less diverse reasoning paths.
>
> - **Results and comparison.**
>
>   Table 2: Comparison of different trajectory selection strategies.
>
>   | Selection Strategy    | Pass@16 (%) |
>   | --------------------- | ----------- |
>   | High-Entropy          | 53.68       |
>   | Low-Entropy           | 48.78       |
>   | Entropy-free (Random) | 50.42       |
>
>   Because all variants use the **same verifier** and differ *only* in whether and how entropy is applied, this controlled comparison directly demonstrates that the gains arise from **entropy-driven exploration**, rather than arbitrary subsampling of correct trajectories.

---

### Official Review · Reviewer_zqwr · 2025-11-10

**Soundness:** 3
**Presentation:** 3
**Contribution:** 2
**Rating:** 4
**Confidence:** 4

**Summary:**

This paper proposes an entropy-aware self-evolution framework that addresses the fundamental trade-off between correctness and exploration in iterative LLM training. The key insight is that high-entropy trajectories (those with verified correctness but intrinsic uncertainty) provide both reliable supervision and signal for exploring alternative reasoning paths. The framework employs two complementary strategies: high-entropy selection that prioritizes verified trajectories with high uncertainty to ensure informative yet dependable training signals, and entropy-aware rethinking that identifies high-uncertainty reasoning steps for truncation and regeneration to uncover diverse solutions. The authors establish theoretical connections between sequence-level entropy and expected supervised fine-tuning loss, demonstrating that high-entropy trajectories induce larger expected gradients and richer learning signals.

**Strengths:**

The paper presents a well-motivated and principled approach to a genuine problem in self-evolution—balancing correctness with exploration. The key insight that high-entropy verified trajectories can serve dual purposes (reliable supervision and exploration signals) is intuitive and well-articulated. The theoretical contribution connecting entropy to expected supervised loss provides solid grounding for the approach. The experimental evaluation is reasonably comprehensive, testing across multiple model sizes and benchmarks with consistent improvements over ENVISIONS baseline. The analysis sections provide valuable insights into why the method works, including visualizations of similarity scores, linguistic patterns in high-entropy tokens, and ablation studies. The framework is clearly described with well-designed figures, and the writing is generally accessible.

**Weaknesses:**

The experiments are limited to relatively small models (1.8B parameters maximum) due to computational constraints, raising questions about scalability and whether the findings generalize to larger LLMs. The paper compares against only one primary baseline (ENVISIONS), making it difficult to assess whether improvements stem from the entropy-aware design specifically or from implementation details. The computational overhead is not systematically analyzed—it's unclear whether the gains justify the additional verification and computation costs. Additionally, while trajectory rethinking is empirically validated, the theoretical justification for why truncating at high-entropy tokens specifically should improve diversity is limited.

**Questions:**

What is the actual computational cost and wall-clock time overhead compared to ENVISIONS, and how does this scale with dataset size? Can the framework work with differentiable or neural verifiers rather than only executable code verification? Why should entropy specifically be the signal for exploration potential rather than other measures of uncertainty or diversity? The theoretical analysis shows entropy relates to expected loss magnitude but doesn't directly justify why this improves exploration, is there a more principled connection? How does the method perform with different sampling temperatures, and could the improvements partly be explained by implicit temperature regularization?

---

> ### Author Response · Authors · 2025-11-21
>
> ### 1. Computational Cost and Scalability
>
> **Response to the question:**
>
> > "What is the actual computational cost and wall-clock time overhead compared to ENVISIONS ? "
>
> - We agree that a systematic cost analysis is important. In the revision, We have added a detailed comparison with ENVISIONS in the **Appendix D** , including **Wall-clock time, FLOPs, and cost per Pass@K improvement**.
>
> - Based on Appendix analyses, we summarize the overall computation, FLOPs, wall-clock time, and cost-performance trade-off of our framework compared to ENVISIONS:
>
>   | Framework                     | Stage       | FLOPs ($\times10^{15}$) | Wall-clock (h) | Notes                                             |
>   | ----------------------------- | ----------- | ----------------------- | -------------- | ------------------------------------------------- |
>   | **Ours ($I$=10, $K$=5)**      | Exploration | 475.0                   | --             | Higher FLOPs in Exploration and SFT               |
>   |                               | Rethinking  | 373.0                   | --             | Reduced intermediate stage cost drives efficiency |
>   |                               | SFT         | 935.5                   | --             |                                                   |
>   |                               | **Total**   | 1783.5                  | 139.3          |                                                   |
>   | **ENVISIONS ($I$=10, $K$=5)** | Exploration | 450.7                   | --             | Higher cost in Refinement stage                   |
>   |                               | Refinement  | 751.9                   | --             |                                                   |
>   |                               | SFT         | 856.6                   | --             |                                                   |
>   | ****                          | **Total**   | 2059.2                  | 152.0          |                                                   |
>
>   **Cost-Performance Trade-off:**
>
> - Our method achieves **lower total FLOPs (1.78×10¹⁸ vs 2.06×10¹⁸)** and **slightly shorter wall-clock time (139.3h vs 152.0h)**.
>
>   - The reduction mainly comes from the **Trajectory Rethinking** stage, while Exploration and SFT stages consume slightly more FLOPs.
>   - Performance gains are significant: e.g., **GSM8K Pass@16 improves 8.27%** over few-shot baseline.
>   - Normalized by FLOPs and wall-clock time, our framework achieves **higher cost-efficiency** than ENVISIONS:
>
> - This demonstrates that our framework provides a favorable **trade-off between additional computation and performance gain**, and addresses the reviewer’s concern about scalability and efficiency.
>
> **Response to the question:**
>
> > "How the computational cost scales with data size ? "
>
> - In both the **exploration** and **rethinking** stages, FLOPs scale linearly with the number of questions, since we fix the per-question inference budget. For the **SFT stage** after selection, we restrict the selected samples to a fixed top-N subset, which ensures that the **upper bound** of training cost also grows at most linearly with the data size.However, for larger-scale self-evolution settings, the model can learn more effectively from the expanded data, and thus the amount of selected positive SFT data tends to approach this upper bound.
>
> **Response to the weakness:**
>
> > "The limitation of experiments to relatively small models (up to 1.8B parameters) and the concerns about scalability to larger LLMs "
>
> - We sincerely appreciate your insightful comment regarding this. Unfortunately, due to limited time and computational resources, we are currently unable to conduct experiments on larger-scale models. We hope to address this in future work and believe our current results still provide meaningful insights into the proposed framework.

---

> ### Author Response · Authors · 2025-11-21
>
> ### 2. Verifier Flexibility
>
> **Response to the question:**
>
> > "Can the framework work with differentiable or neural verifiers instead of only executable code verification?"
>
> - We thank the reviewer for this suggestion. Our framework is **not tightly coupled to the verifier type**: while we used executable-code verification for strong correctness guarantees, the core mechanisms of *high-entropy selection* and *entropy-aware rethinking* only require a reliable verification signal. In principle, a **neural or differentiable verifier** could replace executable code.
> - However, executable verification provides an additional advantage that is crucial for reasoning: **it ensures that every intermediate step in the trajectory is concretely verifiable**. This prevents situations where an incorrect or logically inconsistent reasoning path is mistakenly judged as correct. In contrast, **neural verifiers may be noisier** and can occasionally assign positive feedback to flawed reasoning, which would weaken the reliability of entropy-based exploration and rethinking.
> - Extending our method to neural verifiers is still promising future work, but doing so would likely require additional stabilization techniques (e.g., verifier calibration or training on verified trajectories) to maintain the fidelity of the reasoning signal.

---

> ### Author Response · Authors · 2025-11-21
>
> ### 3. Entropy as an Exploration Signal
>
> **Response to the question:**
>
> > "Why should entropy specifically be the signal for exploration potential rather than other measures of uncertainty or diversity? "
>
> - We use **sequence-level Shannon entropy** because it is a model-native and computationally stable measure that directly reflects the uncertainty of the model’s predictive distribution and admits a closed-form connection to the expected supervised loss used in SFT (Eq.(9)–(12)). This makes entropy an efficient proxy for how strongly a sampled trajectory can influence parameter updates.
>   Importantly, we do *not* claim entropy is the only possible signal. Other uncertainty measures such as **mutual information** [1] or **epistemic–aleatoric decomposition** [2] are also principled. Our method works well with entropy primarily because we pair it with a **validator**, which effectively filters out aleatoric noise. This makes entropy behave more like an epistemic-uncertainty indicator in practice, enabling useful exploration rather than amplifying noise.
>
> **Response to the question:**
>
> > "The theoretical analysis shows entropy relates to expected loss magnitude but doesn't directly justify why this improves exploration, is there a more principled connection? "
>
> - We thank the reviewer for raising this important point. The rationale for why entropy-based selection promotes effective exploration can be understood from two complementary theoretical perspectives:
>
>   1. **Bayesian Active Learning Perspective:**
>      In Bayesian active learning, predictive entropy$H[y|x,D]$ provides an upper bound on the mutual information between the model parameters $\theta$ and the labels $y$ [1]:
>
>      $$I[y;θ∣x,D]=H[y∣x,D]−E_{θ∼p(θ∣D)}[H[y∣x,θ]]≤H[y∣x,D].$$
>
>      Selecting high-entropy samples thus targets points with potentially higher information gain, effectively reducing epistemic uncertainty.
>
>   2. **Maximum-Entropy Reinforcement Learning Perspective:**
>      In maximum-entropy RL [3, 4], an entropy term is added to the reward to encourage broader behavioral coverage and prevent premature collapse into overconfident modes:
>
>      $π∗=arg⁡max⁡πE_{τ∼π}[\sum^T_{t=0}r(s_t,a_t)+αH(π(⋅∣s_t))].$
>
>      Similarly, learning from verified high-entropy trajectories encourages the model to explore beyond its current confident reasoning paths.
>
>   Together, these perspectives justify that high-entropy selection not only amplifies gradient signals (via expected supervised loss) but also facilitates structured exploration, making entropy a principled and effective proxy for exploration in our framework.
>
>   In the revision, we will clarify this in detail and present the discussion in **Appendix E.1-4**.
>
> **Response to the weakness:**
>
> > "Additionally, while trajectory rethinking is empirically validated, the theoretical justification for why truncating at high-entropy tokens specifically should improve diversity is limited."
>
> Thank you for raising this point. We have added a concise theoretical justification in **Appendix E.5**.
> Intuitively, high-entropy tokens correspond to decision forks in a reasoning trajectory: positions where the model’s predictive distribution exhibits the largest branching factor and thus determines which reasoning path the model will follow. Formally, the local branching factor scales as $exp(H_t)$ so diversity in reachable trajectories is dominated by the few positions with high entropy. Re-sampling at these positions yields a multiplicative increase in trajectory diversity per unit compute, whereas resampling at near-deterministic tokens contributes negligibly.
>
> *References:*
>
> [1] Houlsby, N., Huszár, F., Ghahramani, Z. and Lengyel, M., 2011. Bayesian active learning for classification and preference learning. arXiv preprint arXiv:1112.5745.
>
> [2] Kendall, A. and Gal, Y., 2017. What uncertainties do we need in Bayesian deep learning for computer vision? Advances in Neural Information Processing Systems, 30.
>
> [3] Ziebart, B.D., 2010. Modeling purposeful adaptive behavior with the principle of maximum causal entropy. Ph.D. dissertation, Carnegie Mellon University.
>
> [4] Haarnoja, T., Zhou, A., Abbeel, P. and Levine, S., 2018. Soft actor-critic: Off-policy maximum entropy deep reinforcement learning with a stochastic actor. In Proceedings of the 35th International Conference on Machine Learning (ICML), pp.1861–1870.

---

> ### Author Response · Authors · 2025-11-21
>
> ### 4. Sampling Temperature and Implicit Regularization
>
> **Response to the question:**
>
> > "How does the method perform with different sampling temperatures, and could improvements partly come from implicit temperature regularization?"
>
> - To directly examine whether the observed gains could be partly attributed to implicit temperature regularization, we tested the trained model on the GSM8K test set under sampling temperatures ranging from 0.6 to 1.2. We find that performance first increases with temperature and then decreases at higher temperatures, confirming that sampling temperature indeed affects output diversity and thus Pass@K performance. However, across wide range of temperatures, our method consistently outperforms ENVISIONS, indicating that the improvements cannot be explained solely by temperature effects.
>
> **Pass@16 accuracy under different sampling temperatures (InternLM2.5-1.8B, 1k-sample training subset)**
>
> | Temperature   | 0.6   | 0.7   | 0.8   | 0.9   | 1.0   | 1.1   | 1.2   |
> | ------------- | ----- | ----- | ----- | ----- | ----- | ----- | ----- |
> | **Ours**      | 53.68 | 54.59 | 57.62 | 57.77 | 59.21 | 58.83 | 53.30 |
> | **ENVISIONS** | 50.27 | 52.62 | 52.77 | 54.21 | 53.91 | 54.06 | 53.37 |

---

### Author Response · Authors · 2025-11-21
**About the Revision of the PDF**

We sincerely thank the reviewers for their constructive suggestions. In response, we have revised the manuscript accordingly and uploaded a diff version of the updated PDF.
All modifications in the main text are highlighted in red, and the newly added appendix sections are marked with red section titles.

---

### Author Response · Authors · 2025-12-03
**Overview of Revisions Addressing Reviewer Feedback**

TL;DR. We propose an entropy‑aware self‑evolution framework to improve reasoning in LLMs by balancing correctness and exploration. In the revised submission, we addressed all major concerns raised by the reviewers via clarified motivation, strengthened theoretical justification, ablation studies, and extended experiments — resolving the critical issues and demonstrating that our method indeed improves both reliability and reasoning diversity.

We summarize the reviewers’ key concerns and our corresponding resolutions as follows:

**Reviewer zqwr**: Concerned the computational cost and theoretical justification of using high-entropy tokens for trajectory rethinking, expressing doubts about whether improvements generalize to larger models or alternative verifiers.

**Resulution**: We provided a detailed computational-cost analysis showing favorable FLOPs and wall-clock trade-offs, and added both Bayesian active learning and maximum-entropy RL perspectives to theoretically justify why high-entropy tokens effectively drive exploration and increase trajectory diversity.

**Reviewer tmXE**: Concerned clearer isolation of each component’s contribution and verification that high-entropy trajectory selection itself drives gains, rather than simply generating more trajectories.

**Resulution**: We conducted compute-matched ablations and added low-entropy and entropy-free (random) selection baselines, showing that both selection and rethinking independently improve performance, and that the full method combining high-entropy selection with rethinking achieves the largest gain, confirming that entropy-driven exploration and the rethink mechanism together—not just more trajectories—account for the improvement.

**Reviewer cuqe**: Concerned the framework’s generalization beyond program-verifiable math, limited evidence in single-shot/small-budget settings, and absence of hyperparameter sensitivity analysis.

**Resulution**: We clarified that the method is compatible with weaker verifiers, demonstrated consistent gains in low-K/single-shot settings showing improved sample efficiency, and provided thorough sensitivity studies with guidelines for all key hyperparameters to ensure reproducibility.

**Reviewer Q3Hi**: Concerned clearer differentiation from ENVISIONS, more comprehensive ablations for the Trajectory Rethinking stage and filtering criteria, and qualitative examples showing how reasoning chains evolve across iterations.

**Resolution**: We clarified the key differences between our method and ENVISIONS, conducted compute-matched and filtering-strategy ablations demonstrating the effectiveness of high-entropy selection and Trajectory Rethinking, and provided qualitative examples in the appendix showing that our approach produces faster, more diverse, and higher-quality reasoning trajectories compared to ENVISIONS.

---

### Meta-Review · Area_Chair_h9D9 · 2026-01-07

**Summary:**

This paper proposes an entropy-aware self-evolution framework for LLM reasoning that explicitly balances correctness and exploration by prioritizing verified high-entropy trajectories and selectively rethinking uncertain reasoning steps. The core idea is theoretically grounded by linking sequence-level entropy to expected supervised fine-tuning loss, arguing that high-entropy trajectories provide stronger learning signals. Experiments across several math reasoning benchmarks show consistent Pass@K improvements over the ENVISIONS baseline, with detailed analyses and ablations explaining where the gains come from. Overall, reviewers saw the method as principled, well-motivated, and empirically solid, while raising concerns about scalability, baselines, cost, and generalization beyond program-verifiable domains.

**Reviewer Concerns:**

Reviewer zqwr’s concerns about computational cost, scalability, entropy justification, verifier flexibility, temperature effects, and the rationale for entropy-based rethinking were comprehensively addressed with new cost analyses, theoretical extensions, temperature studies, and clarifications, while the limitation to small models remains acknowledged but unresolved.

Reviewer tmXE’s requests for clearer component-level ablations were fully addressed through compute-matched ablations, entropy-vs-random comparisons, and added analyses isolating selection and rethinking contributions.

Reviewer cuqe’s concerns about cost reporting, low-budget performance, hyperparameter sensitivity, and reproducibility were largely resolved with detailed cost-efficiency tables, low-K results, and sensitivity discussions, but their request for empirical validation on semantic or open-ended reasoning with LLM judges remains explicitly unaddressed and deferred to future work.

Reviewer Q3Hi’s concerns about limited baselines, unclear differences from ENVISIONS, narrow ablations, and lack of qualitative trajectory evolution were substantially addressed through expanded comparisons, clarified methodological distinctions, richer ablations, and added qualitative examples, though the critique about benchmark datedness is acknowledged rather than resolved.

**Reviewer Scores:**

Reviewer zqwr (4) did not explicitly state a score update, but since all technical questions were addressed except scalability to larger models, the rebuttal likely partially but not fully resolves their concerns.

Reviewer tmXE (6) did not comment on changing their score, but given that all requested ablations and baselines were added, their concerns appear fully addressed.

Reviewer cuqe (6/5) explicitly maintained a remaining concern after the rebuttal regarding the lack of semantic/open-ended experiments, indicating their questions were not fully resolved.

Reviewer Q3Hi (4) did not state a score update, but methodological and ablation-related concerns were directly answered and expanded in the revision. However, some other question regarding the motivation and setups are not fully addressed.

---

### Decision · Program_Chairs · 2026-01-26

Reject